# RLSbench: A Large-Scale Empirical Study of Domain Adaptation Under Relaxed Label Shift

## Abstract

Despite the emergence of principled methods for domain adaptation under label shift (where only the class balance changes), the sensitivity of these methods to natural-seeming covariate shifts remains precariously underexplored. Meanwhile, popular deep domain adaptation heuristics, despite showing promise on benchmark datasets, tend to falter when faced with shifts in the class balance. Moreover, it's difficult to assess the state of the field owing to inconsistencies among relevant papers in evaluation criteria, datasets, and baselines. In this paper, we introduce RLSbench, a large-scale benchmark for such *relaxed label shift* settings, consisting of 14 datasets across vision, tabular, and language modalities spanning >500 distribution shift pairs with different class proportions. We evaluate 13 popular domain adaptation methods, demonstrating a more widespread susceptibility to failure under extreme shifts in the class proportions than was previously known. We develop an effective meta-algorithm, compatible with most deep domain adaptation heuristics, that consists of the following two steps: (i) *pseudo-balance* the data at each epoch; and (ii) adjust the final classifier with (an estimate of) target label distribution. In our benchmark, the meta-algorithm improves existing domain adaptation heuristics often by 2–10% accuracy points when label distribution shifts are extreme and has minimal (i.e., <0.5%) to no effect on accuracy in cases with no shift in label distribution. We hope that these findings and the availability of RLSbench will encourage researchers to rigorously evaluate proposed methods in relaxed label shift settings. Code is publicly available at https://github.com/ICLR2023Anon.

## 1 Introduction

Real-world deployments of machine learning models are typically characterized by distribution shift, where data encountered in production exhibits statistical differences from the available training data (Quinonero-Candela et al., 2008; Torralba & Efros, 2011; Koh et al., 2021). Because continually labeling data can be prohibitively expensive, researchers have focused on the unsupervised domain adaptation (DA) setting, where only labeled data sampled from the *source* distribution and unlabeled from the *target* distribution are available for training.

Absent further assumptions, the DA problem is well known to be underspecified (Ben-David et al., 2010b) and thus no method is universally applicable. Researchers have responded to these challenges in several ways. One approach is to investigate additional assumptions that render the problem well-posed. Popular examples include covariate shift and label shift, for which identification strategies and principled methods exist whenever the source and target distributions have overlapping support (Shimodaira, 2000; Schölkopf et al., 2012; Gretton et al., 2009). Under label shift in particular, recent research has produced effective methods that are applicable in deep learning regimes and yield both consistent estimates of the target label marginal and principled ways to update the resulting classifier (Lipton et al., 2018; Alexandari et al., 2021; Azizzadenesheli et al., 2019; Garg et al., 2020). However, these assumptions are typically, to some degree, violated in practice. Even for archetypal cases like shift in disease prevalence (Lipton et al., 2018), the label shift assumption can be violated. For example, over the course of the COVID-19 epidemic, changes in disease positivity have been coupled with shifts in the age distribution of the infected and subtle mutations of the virus itself.

A complementary line of research focuses on constructing benchmark datasets for evaluating methods, in the hopes of finding heuristics that, for the kinds of problems that arise in practice, tend

to incorporate the unlabeled target data profitably. Examples of such benchmarks include Office-Home (Venkateswara et al., 2017), Domainnet (Peng et al., 2019)), WILDS (Sagawa et al., 2021). However, most academic benchmarks exhibit little or no shift in the label distribution $p(y)$. Consequently, benchmark-driven research has produced a variety of heuristic methods (Ganin et al., 2016; Sohn et al., 2020; Wang et al., 2021; Li et al., 2016) that despite yielding gains in benchmark performance tend to break when $p(y)$ shifts. While this has previously been shown for domain-adversarial methods (Wu et al., 2019; Zhao et al., 2019), we show that this problem is more widespread than previously known. Several recent papers attempt to address shift in label distribution compounded by natural variations in $p(x|y)$ (Tan et al., 2020; Tachet des Combes et al., 2020; Prabhu et al., 2021). However, the experimental evaluations are hard to compare across papers owing to discrepancies in how shifts in $p(y)$ are simulated and the choice of evaluation metrics. Moreover, many methods violate the unsupervised contract by peeking at target validation performance during model selection and hyperparameter tuning. In short, there is a paucity of comprehensive and fair comparisons between DA methods for settings with shifts in label distribution.

In this paper, we develop RLSBENCH, a standarized test bed of *relaxed label shift* settings, where $p(y)$ can shift arbitrarily and the class conditionals $p(x|y)$ can shift in seemingly natural ways (following the popular DA benchmarks). We evaluate a collection of popular DA methods based on domain-invariant representation learning, self-training, and test-time adaptation across 14 multi-domain datasets spanning vision, Natural Language Processing (NLP), and tabular modalities. The different domains in each dataset present a different shift in $p(x|y)$. Since these datasets exhibit minor to no shift in label marginal, we simulate shift in target label marginal via stratified sampling with varying severity. Overall, we obtain 560 different source and target distribution shift pairs and train $> 30k$ models in our testbed.

Based on our experiments on RLSBENCH suite, we make several findings. First, we observe that while popular DA methods often improve over a source-only classifier absent shift in target label distribution, their performance tends to degrade, dropping below source-only classifiers under severe shifts in target label marginal. Next, we develop a meta-algorithm with two simple corrections: (i) re-sampling the data to balance the source and pseudo-balance the target; (ii) re-weighting the final classifier using an estimate of the target label marginal. We observe that in these relaxed label shift environments, the performance of existing DA methods (e.g. CDANN, FixMatch, and BN-adapt) when paired with our meta-algorithm tends to improve over source-only classifier. On the other hand, existing methods specifically proposed for relaxed label shift (e.g., IW-CDANN and SENTRY) often fail to improve over a source-only classifier and significantly underperform when compared to existing DA methods paired with our meta-algorithm.

Our findings underscore the importance of a fair comparison to avoid a false sense of scientific progress in relaxed label shift scenarios. Moreover, we hope that the RLSBENCH testbed and our meta-algorithm (that can be paired with any DA method) provide a framework for rigorous and reproducible future research in relaxed label shift scenarios.

## 2 PRELIMINARIES AND PRIOR WORK

We first setup the notation and formally define the problem. Let $\mathcal{X}$ be the input space and $\mathcal{Y} = \{1, 2, \ldots, k\}$ the output space. Let $P_s, P_t : \mathcal{X} \times \mathcal{Y} \to [0, 1]$ be the source and target distributions and let $p_s$ and $p_t$ denote the corresponding probability density (or mass) functions. Unlike the standard supervised setting, in unsupervised DA, we possess labeled source data $\{(x_1, y_1), (x_2, y_2), \ldots, (x_n, y_n)\}$ and unlabeled target data $\{x_{n+1}, x_{n+2}, \ldots, x_{n+m}\}$. With $f : \mathcal{X} \to \Delta^{k-1}$, we denote a predictor function which predicts $\widehat{y} = \arg\max_y f_y(x)$ on an input $x$. For a vector $v$, we use $v_y$ to access the element at index $y$.

In the traditional label shift setting, one assumes that $p(x|y)$ does not change but that $p(y)$ can. Under label shift, two challenges arise: (i) estimate the target label marginal $p_t(y)$; and (ii) train a classifier $f$ to maximize the performance on target domain. This paper focuses on the *relaxed label shift* setting. In particular, we assume that the label distribution can shift from source to target arbitrarily but that $p(x|y)$ varies between source and target in some comparatively restrictive way (e.g., shifts arising naturally in the real-world like ImageNet (Russakovsky et al., 2015) to ImageNetV2 (Recht et al., 2019)). Mathematically, we assume a divergence-based restriction on $p(x|y)$. That is, for some small $\epsilon > 0$ and distributional distance $\mathcal{D}$, we have $\max_y \mathcal{D}(p_s(x|y), p_t(x|y)) \leqslant \epsilon$ and allow an arbitrary shift in the label marginal $p(y)$. We discuss several precise instantiations in App. F. However, in

practice, it's hard to empirically verify these distribution distances for small enough $\epsilon$ with finite samples. Moreover, we lack a rigorous characterization of the sense in which those shifts arise in popular DA benchmarks, and since, the focus of our work is on the empirical evaluation with real-world datasets, we leave a formal investigation for future work. While prior work addressing relaxed label shift has primarily focused on classifier performance, we also separately evaluate methods for estimating the target label marginal. This can be beneficial for two reasons. First, it can shed more light into how improving the estimates of target class proportion improves target performance. Second, understanding how the class proportions are changing can be of independent interest.

## 2.1 PRIOR WORK

**Unsupervised domain adaption** In our work, we focus on unsupervised DA where the goal is to adapt a predictor from a source distribution with labeled data to a target distribution from which we only observe unlabeled examples. Two popular settings for which DA is well-posed include (i) *covariate shift* (Zhang et al., 2013; Zadrozny, 2004; Cortes et al., 2010; Cortes & Mohri, 2014; Gretton et al., 2009) where $p(x)$ can change from source to target but $p(y|x)$ remains invariant; and (ii) *label shift* (Saerens et al., 2002; Lipton et al., 2018; Azizzadenesheli et al., 2019; Alexandari et al., 2021; Garg et al., 2020; Zhang et al., 2021) where the label marginal $p(y)$ can change but $p(x|y)$ is shared across source and target. Principled methods with strong theoretical guarantees exists for adaptation under these settings when target distribution's support is a subset of the source support. Ben-David et al. (2010b;a); Mansour et al. (2009); Zhao et al. (2019); Wu et al. (2019) present theoretical analysis when the assumptions of contained support is violated. More recently, a massive literature has emerged exploring a benchmark-driven heuristic approach (Long et al., 2015; 2017; Sun & Saenko, 2016; Sun et al., 2017; Zhang et al., 2019; 2018; Ganin et al., 2016; Sohn et al., 2020). However, rigorous evaluation of popular DA methods is typically restricted to these carefully curated benchmark datasets where their is minor to no shift in label marginal from source to target.

**Relaxed Label Shift** Exploring the problem of shift in label marginal from source to target with natural variations in $p(x|y)$, a few papers highlighted theoretical and empirical failures of DA methods based on domain-adversarial neural network training (Yan et al., 2017; Wu et al., 2019; Zhao et al., 2019). Subsequently, several papers attempted to handle these problems in domain-adversarial training (Tachet et al., 2020; Prabhu et al., 2021; Liu et al., 2021; Tan et al., 2020; Manders et al., 2019). However, these methods often lack comparisons with other prominent DA methods and are evaluated under different datasets and model selection criteria. To this end, we perform a large scale rigorous comparison of prominent representative DA methods in a standardized evaluation framework.

**Domain generalization** In domain generalization, the model is given access to data from multiple different domains and the goal is to generalize to a previously unseen domain at test time (Blanchard et al., 2011; Muandet et al., 2013). For a survey of different algorithms for domain generalization, we refer the reader to Gulrajani & Lopez-Paz (2020). A crucial distinction here is that unlike the domain generalization setting, in DA problems, we have access to unlabeled examples from the test domain.

**Distinction from previous distribution shift benchmark studies** Previous studies evaluating robustness under distribution shift predominantly focuses on transfer learning and domain generalization settings Wenzel et al. (2022); Gulrajani & Lopez-Paz (2020); Djolonga et al. (2021); Wiles et al. (2021); Koh et al. (2021). Taori et al. (2020); Hendrycks et al. (2021) studies the impact of robustness interventions (e.g. data augmentation techniques, adversarial training) on target (out of distribution) performance. Notably, Sagawa et al. (2021) focused on evaluating DA methods on WILDS-2.0, an extended WILDS benchmark for DA setting. Our work is complementary to these studies, as we present the first extensive study of DA methods under shift in $p(y)$ and natural variations in $p(x|y)$.

## 3 RLSBENCH: A BENCHMARK FOR RELAXED LABEL SHIFT

In this section, we introduce RLSBENCH, a suite of datasets and DA algorithms that are at the core of our study. Motivated by correction methods for the (stricter) label shift setting (Saerens et al., 2002; Lipton et al., 2018) and learning under imbalanced datasets (Wei et al., 2021; Cao et al., 2019a), we also present a meta-algorithm with simple corrections compatible with almost any DA method.

### 3.1 DATASETS

RLSBENCH builds on fourteen open-source multi-domain datasets for classification. We include tasks spanning applications in object classification, satellite imagery, medicine, and toxicity detection:

(i) **CIFAR-10** which includes the original CIFAR-10 (Krizhevsky & Hinton, 2009), CIFAR-10-C (Hendrycks & Dietterich, 2019) and CIFAR-10v2 (Recht et al., 2018); (ii) **CIFAR-100** including the original dataset and CIFAR-100-C; (iii) all four BREEDs datasets (Santurkar et al., 2021), i.e., **Entity13**, **Entity30**, **Nonliving26**, **Living17**. BREEDs leverages class hierarchy in ImageNet (Russakovsky et al., 2015) to repurpose original classes to be the subpopulations and define a classification task on superclasses. We consider distribution shift due to subpopulation shift which is induced by directly making the subpopulations present in the source and target distributions disjoint. We also consider natural shifts induced due to differences in the data collection process of ImageNet, i.e, ImageNetv2 (Recht et al., 2019) and a combination of both. (iv) **OfficeHome** (Venkateswara et al., 2017) which includes four domains: art, clipart, product, and real; (v) **DomainNet** (Peng et al., 2019) where we consider four domains: clipart, painting, real, sketch; (vi) **Visda** (Peng et al., 2018) which contains three domains: train, val and test; (vii) **FMoW** (Koh et al., 2021; Christie et al., 2018) from WILDS benchmark which includes three domains: train, OOD val, and OOD test—with satellite images taken in different geographical regions and at different times; (viii) **Camelyon** (Bandi et al., 2018) from WILDS benchmark which includes three domains: train, OOD val, and OOD test, for tumor identification with domains corresponding to different hospitals; (ix) **Civilcomments** (Borkan et al., 2019) which includes three domains: train, OOD val, and OOD test, for toxicity detection with domains corresponding to different demographic subpopulations; (x) **Retiring Adults** (Ding et al., 2021) where we consider the ACSIncome prediction task with various domains representing different states and time-period; and (xi) **Mimic Readmission** (Johnson et al., 2020; PhysioBank, 2000) where the task is to predict readmission risk with various domains representing data from different time-period.

Throughout the paper, we represent each multi-domain dataset with the name highlighted in the boldface above. Across these datasets, we obtain a total of 56 different source and target pairs. We relegate other details about datasets in App. D. For vision datasets, we show example images in Fig. 7.

**Simulating a shift in target marginal**  The above datasets present minor to no shift in label marginal. Hence, we simulate such a shift by altering the target label marginal and keeping the source target distribution fixed (to the original source label distribution). Note that, unlike some previous studies, we do not alter the source label marginal because, in practice, we may have an option to carefully curate the training distribution but might have little to no control over the test data.

For each target dataset, we have the true labels which allow us to vary the target label distribution. In particular, we sample the target label marginal from a Dirichlet distribution with a parameter $\alpha \in \{0.5, 1, 3.0, 10\}$ multiplier to the original target marginal. Specifically, $p_t(y) \sim \text{Dir}(\beta)$ where $\beta_y = \alpha \cdot p_{t,0}(y)$ and $p_{t,0}(y)$ is the original target label marginal. The Dirichlet parameter $\alpha$ controls the severity of shift in target label marginal. Intuitively, as $\alpha$ decreases, the severity of the shift increases. For completeness, we also include the target dataset with the original target label marginal. For ease of exposition, we denote the shifts as NONE (no external shift) in the set of Dirichlet parameters, i.e. the limiting distribution as $\alpha \to \infty$. After simulating the shift in the target label marginal (with two seeds for each $\alpha$), we obtain 560 pairs of different source and target datasets.

### 3.2 DOMAIN ADAPTATION METHODS

With the current version of RLSBENCH, we implement the following algorithms (a more detailed description of each method is included in App. K):

**Source only**  As a baseline, we include model trained with empirical risk minimization (Vapnik, 1999) with cross-entropy loss on the source domain. We include source only models trained with and without augmentations. We also include adversarial robust models trained on source data with augmentations (**Source (adv)**). In particular, we use models adversarially trained against $\ell_2$-perturbations.

**Domain alignment methods**  These methods employ domain-adversarial training schemes aimed to learn invariant representations across different domains (Ganin et al., 2016; Zhang et al., 2019; Tan et al., 2020). For our experiments, we include the following *five* methods: Domain Adversarial Neural Networks (**DANN** (Ganin et al., 2016)), Conditional DANN (**CDANN** (Long et al., 2018), Maximum Classifier Discrepancy (**MCD** (Saito et al., 2018)), Importance-reweighted DANN and CDANN (i.e. **IW-DANN** & **IW-CDANN** Tachet des Combes et al. (2020)).

**Self-training methods**  These methods "pseudo-label" unlabeled examples with the model's own predictions and then train on them as if they were labeled examples. For vision datasets, these methods often also use consistency regularization, which encourages the model to make consistent predictions

---

**Algorithm 1** Meta algorithm to handle shift in class proportions

---

**input** Source training and validation data: $(X_S, Y_S)$ and $(X'_S, Y'_S)$, unlabeled target training and validation data: $X_T$ and $X'_T$, classifier $f$, and DA algorithm $\mathcal{A}$

1: $\widetilde{X}_S, \widetilde{Y}_S \leftarrow \text{SampleClassBalanced}(X_S, Y_S)$          ▷ `Balance source data`
2: **for** $t = 1$ to $T$ **do**
3:     $\widehat{Y}_T \leftarrow \arg\max_y f_y(X_T)$
4:     $\widetilde{X}_T \leftarrow \text{SampleClassBalanced}(X_T, \widehat{Y}_T)$       ▷ `Pseudo-balance target data`
5:     Run an epoch of $\mathcal{A}$ to update $f$ on balanced source data $\{\widetilde{X}_S, \widetilde{Y}_S\}$ and target samples $\{\widetilde{X}_T\}$
6: **end for**
7: Estimate target marginal $\widehat{p}_t(y) \leftarrow \text{EstimateLabelMarginal}(f, X'_S, Y'_S, X'_T)$
8: $f'_j \leftarrow \dfrac{\widehat{p}_t(y = j) \cdot f_j}{\sum_k \widehat{p}_t(y = k) \cdot f_k}$ for all $j \in \mathcal{Y}$
                ▷ `Re-weight predictor with estimated label marginal`

**output** Target label marginal $\widehat{p}_t(y)$ and classifier $f'$

---

on augmented views of unlabeled examples (Lee et al., 2013; Xie et al., 2020; Berthelot et al., 2021). We include the following three algorithms: **FixMatch** (Sohn et al., 2020), **Noisy Student** (Xie et al., 2020), Selective Entropy Optimization via Committee Consistency (**SENTRY** (Prabhu et al., 2021)). For NLP and tabular dataset, where we do not have strong augmentations defined, we consider **PseudoLabel** algorithm (Lee et al., 2013).

**Test-time adaptation methods** These methods take a source-trained model and adapt a few parameters (e.g. batch norm parameters, batch norm statistics) on the unlabeled target data with an aim to improve target performance. We include the following methods: **CORAL** (Sun et al., 2016) or Domain Adjusted Regression (DARE (Rosenfeld et al., 2022)), BatchNorm adaptation (**BN-adapt** (Li et al., 2016; Schneider et al., 2020)), Test entropy minimization (**TENT** (Wang et al., 2021)).

### 3.3 META ALGORITHM TO HANDLE SHIFTS IN TARGET CLASS PROPORTIONS

Here we discuss two simple general-purpose corrections that we implement in our framework. First, note that, as the severity of shift in the target label marginal increases, the performance of DA methods can falter as the training is done over source and target datasets with different class proportions. Indeed, failure of domain adversarial training methods (one category of deep DA methods) has been theoretically and empirically shown in the literature (Wu et al., 2019; Zhao et al., 2019). In our experiments, we show that a failure due to a shift in label distribution is not limited to domain adversarial training methods, but is common with all the popular DA methods (Sec. 4).

**Re-sampling** To handle label imbalance in standard supervised learning, re-sampling the data to balance the class marginal is a known successful strategy (Chawla et al., 2002; Buda et al., 2018; Cao et al., 2019b). In relaxed label shift, we seek to handle the imbalance in the target data (with respect to the source label marginal), where we do not have access to true labels. We adopt an alternative strategy of leveraging pseudolabels for target data to perform pseudo class-balanced re-sampling[1] (Zou et al., 2018; Wei et al., 2021). For relaxed label shift problems, (Prabhu et al., 2021) employed this technique with their committee consistency objective, SENTRY. However, they did not explore re-sampling based correction for existing DA techniques. Since this technique can be used in conjunction with any DA methods, we employ this re-sampling technique with existing DA methods and find that re-sampling benefits all DA methods, often improving over SENTRY in our testbed (Sec. 4).

**Re-weighting** With re-sampling, we can hope to train the classifier $\widehat{f}$ on a mixture of balanced source and balanced target datasets in an ideal case. However, this still leaves open the problem of adapting the classifier $\widehat{f}$ to the original target label distribution which is not available. If we can estimate the target label marginal, we can post-hoc adapt the classifier $\widehat{f}$ with a simple re-weighting correction (Lipton et al., 2018; Alexandari et al., 2021). To estimate the target label marginal, we turn to techniques developed under the stricter label shift assumption (recall, the setting where $p(x|y)$ remains domain invariant). These approaches leverage off-the-shelf classifiers to estimate

---

[1]A different strategy here could be to re-sample target pseudolabel marginal to match source label marginal. For simplicity, in our work, we choose to balance source label marginal and balance target pseudolabel marginal.

target marginal and provide $\mathcal{O}(1/\sqrt{n})$ convergence rates under the label shift condition with mild assumptions on the classifier (Lipton et al., 2018; Azizzadenesheli et al., 2019; Garg et al., 2020).

While the relaxed label shift scenario violates the conditions required for consistency of label shift estimation techniques, we nonetheless employ these techniques and empirically evaluate efficacy of these methods in our testbed. In particular, to estimate the target label marginal, we experiment with: (i) RLLS (Azizzadenesheli et al., 2019); (ii) MLLS (Alexandari et al., 2021); and (iii) *baseline estimator* that simply averages the prediction of a classifier $f$ on unlabeled target data. We provide precise details about these methods in App. E. Since these methods leverage off-the-shelf classifiers, classifiers obtained with any DA methods can be used in conjunction with these estimation methods.

**Summary**  Overall, in Algorithm 1, we illustrate how to incorporate the re-sampling and re-weighting correction with existing DA techniques. Algorithm $\mathcal{A}$ can be any DA method and in Step 7, we can use any of the three methods listed above to estimate the target label marginal. We instantiate Algorithm 1 with several algorithms from Sec. 3.2 in App. K. Intuitively, in an ideal scenario when the re-sampling step in our meta-algorithm perfectly corrects for label imbalance between source and target, we expect DA methods to adapt classifier $f$ to $p(x|y)$ shift. The re-weighting step in our meta-algorithm can then adapt the classifier $f$ to the target label marginal $p_t(y)$. We emphasize that in our work, we *do not* claim to propose these corrections. But, to the best of our knowledge, our work is the first to combine these two corrections together in relaxed label shift scenarios and perform extensive experiments across diverse datasets.

### 3.4 OTHER CHOICES FOR REALISTIC EVALUATION

For a fair evaluation and comparison across different datasets and domain adaptation algorithms, we re-implemented all the algorithms with consistent design choices whenever applicable. We also make several additional implementation choices, described below. We defer the additional details to App. L.

**Model selection criteria and hyperparameter choices**  Given that we lack validation i.i.d data from the target distribution, model selection in DA problems *can not* follow the standard workflow used in supervised training. Prior works often omit details on how to choose hyperparameters leaving open a possibility of choosing hyperparameters using the test set which can provide a false and unreliable sense of improvement. Moreover, inconsistent hyperparameter selection strategies can complicate fair evaluations mis-associating the improvements to the algorithm under study.

In our work, we use source hold-out performance to pick the best hyperparameters. First, for $\ell_2$ regularization and learning rate, we perform a sweep over random hyperparameters to maximize the performance of source only model on the hold-out source data. Then for each dataset, we keep these hyperparameters fixed across DA algorithms. For DA methods specific hyperparameters, we use the same hyperparameters across all the methods incorporating the suggestions made in corresponding papers. Within a run, we use hold out performance on the source to pick the early stopping point. In appendices, we report *oracle* performance by choosing the early stopping point with target accuracy.

**Evaluation criteria**  To evaluate the target label marginal estimation, we report $\ell_1$ error between the estimated label distribution and true target label distribution. To evaluate the classifier performance on target data, we report performance of the (adapted) classifier on a hold-out partition of target data.

**Architectural and pretraining details**  We experiment with different architectures (e.g., DenseNet121, Resenet18, Resnet50, DistilBERT, MLP and Transformer) across different datasets. We experiment with randomly-initialized models and Imagenet, and DistillBert pre-trained models. Given a dataset, we use the same architecture across different DA algorithms.

**Data augmentation**  Data augmentation is a standard ingredient to train vision models which can help approximate some of the variations between domains. Unless stated otherwise, we train all the vision datasets using the standard strong augmentation technique: random horizontal flips, random crops, augmentation with Cutout (DeVries & Taylor, 2017), and RandAugment (Cubuk et al., 2020). To understand help with data augmentations alone, we also experiment with source-only models trained without any data augmentation. For tabular and NLP datasets, we do not use any augmentations.

## 4 MAIN RESULTS

We present aggregated results on vision datasets in our testbed in Fig. 1. In App. B, we present aggregated results on NLP and tabular datasets. We include results on each dataset in App. I. Note that we do not include RS results with a source only model as it is trained only on source data and we

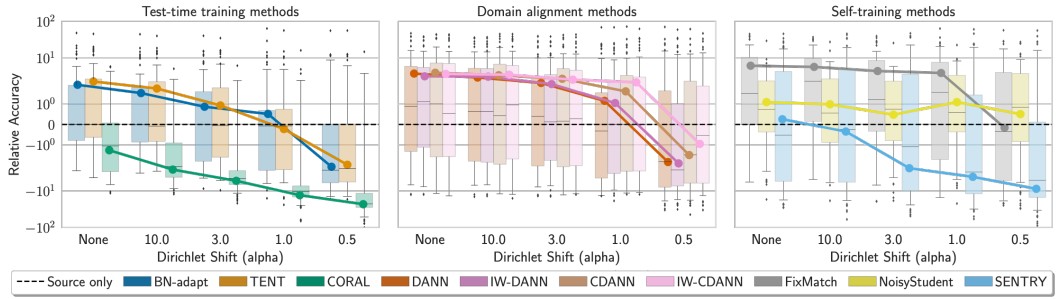

(a) Performance of DA methods relative to source-only training with increasing target label marginal shift

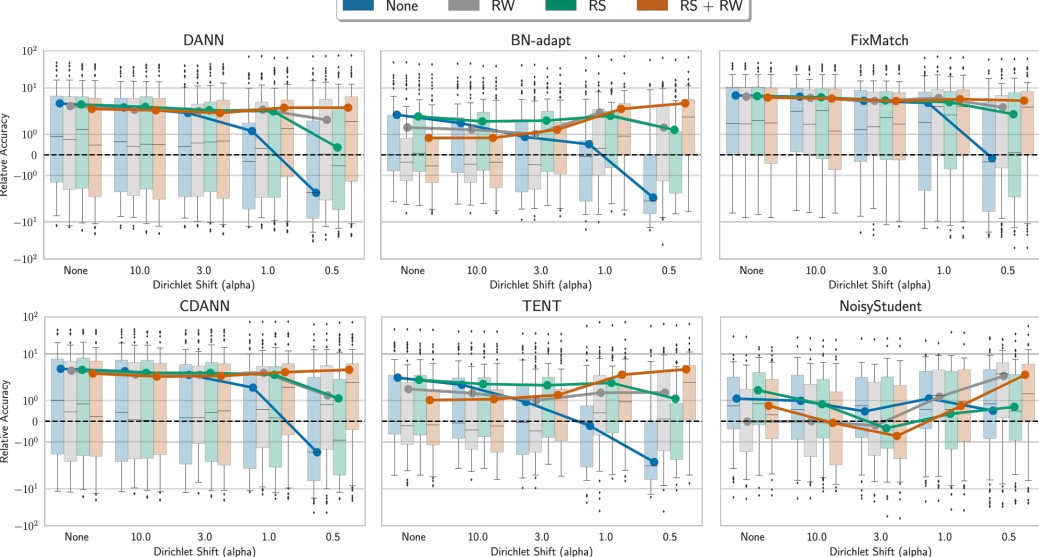

(b) Relative performance of DA methods when paired with our meta-algorithm (RS and RW corrections)

Figure 1: *Performance of different DA methods relative to a source-only model across all distribution shift pairs in vision datasets grouped by shift severity in label marginal.* For each distribution shift pair and DA method, we plot the relative accuracy of the model trained with that DA method by subtracting the accuracy of the source-only model. Hence, the black dotted line at 0 captures the performance of the source-only model. Smaller the Dirichlet shift parameter, the more severe is the shift in target class proportion. **(a)** Shifts with $\alpha = \{\text{NONE}, 10.0, 3.0\}$ have little to no impact on different DA methods whereas the performance of all DA methods degrades when $\alpha \in \{1.0, 0.5\}$ often falling below the performance of a source-only classifier (except for Noisy Student). **(b)** RS and RW (in our meta-algorithm) together significantly improve aggregate performance over no correction for all DA methods. While RS consistently helps (over no correction) across different label marginal shift severities, RW hurts slightly for BN-adapt, TENT, and NoisyStudent when shift severity is small. However, for severe shifts ($\alpha \in \{3.0, 1.0, 0.5\}$) RW significantly improves performance for all the methods. Parallel results on tabular and language datasets in App. B. Detailed results with all methods on individual datasets in App. I. A more detailed description of the plotting technique in App. A.

observed no differences with just balancing the source data (as for most datasets source is already balanced) in our experiments. Unless specified otherwise, we use source validation performance as the early stopping criterion. Based on running our entire RLSBENCH suite, we distill our findings into the following takeaways.

**Popular deep DA methods without any correction falter.** While DA methods often improve over a source-only classifier for cases when the shift in target label marginal is absent or low, the performance of these methods (except Noisy Student) drops below the performance of a source-only classifier when the shift in target label marginal is severe (i.e., when $\alpha = 0.5$ in Fig. 1a, 4a, and 5a). On the other hand, DA methods when paired with RS and RW correction, significantly improve over a source-only model even when the shift in target label marginal is severe (Fig. 1b, 4b, and 5b).

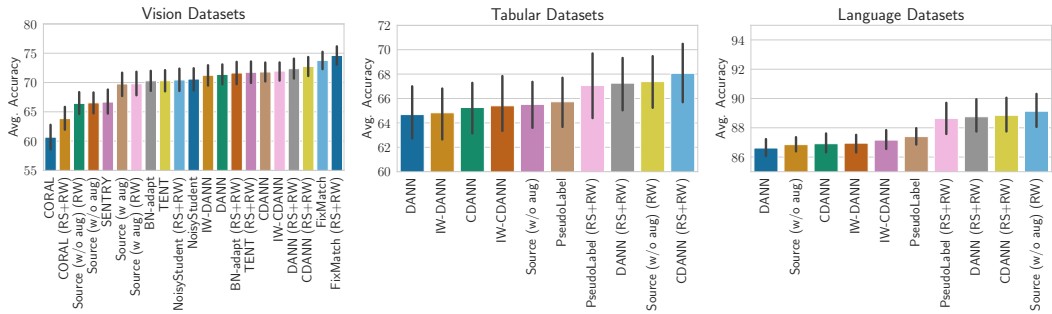

Figure 2: *Average accuracy of different DA methods aggregated across all distribution pairs in each modality.* Parallel results with all methods on individual datasets in App. I.

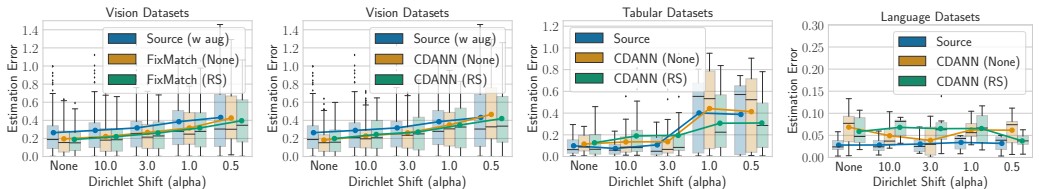

Figure 3: *Target label marginal estimation ($\ell_1$) error with RLLS and classifiers obtained with different DA methods.* Across all shift severities in vision datasets, RLLS with classifiers obtained with DA methods improves over RLLS with a source-only classifier. For tabular datasets, RLLS with classifiers obtained with DA methods improves over RLLS with a source-only classifier for severe target label marginal shifts. Conversely, for NLP datasets, RLLS with source-only classifiers performs better than RLLS with classifiers obtained with DA methods.

**Re-sampling to pseudobalance target often helps all DA methods across all modalities.** When the shift in target label marginal is absent or very small (i.e., $\alpha \in \{\text{NONE}, 10.0\}$ in Fig. 1b, 4b, and 5b), we observe no (significant) differences in performance with re-sampling. However, as the shift severity in target label marginal increases (i.e., $\alpha \in \{3.0, 1.0, 0.5\}$ in Fig. 1b, 4b, and 5b), we observe that re-sampling typically improves all DA methods in our testbed.

**Benefits of post-hoc re-weighting of the classifier depends on shift severity and the underlying DA algorithm.** For domain alignment methods (i.e. DANN and CDANN) and self-training methods, in particular FixMatch and PseudoLabel, we observe that RW correction typically improves (over no correction) significantly when the target label marginal shift is severe (i.e., $\alpha \in \{3.0, 1.0, 0.5\}$ in Fig. 1b, 4b, and 5b) and has no (significant) effect when the shift in target label marginal is absent or very small (i.e., $\alpha \in \{\text{NONE}, 10.0\}$ in Fig. 1b, 4b, and 5b). For BN-adapt, TENT, and NoisyStudent, RW correction can slightly hurt when target label marginal shift is absent or low (i.e., $\alpha \in \{\text{NONE}, 10.0\}$ in Fig. 1b) but continues to improve significantly when the target label marginal shift is severe (i.e., $\alpha \in \{3.0, 1.0, 0.5\}$ in Fig. 1b). Additionally, we observe that in specific scenarios of the real-world shift in $p(x|y)$ (e.g., subpopulation shift in BREEDs datasets, camelyon shifts, and replication study in CIFAR-10 which are benign relative to other vision dataset shifts in our testbed), RW correction does no harm to performance for BN-adapt, TENT, and NoisyStudent even when the target label marginal shift is less severe or absent (refer to datasets in App. I).

**DA methods paired with our meta-algorithm often improve over source-only classifier but no one method consistently performs the best.** First, we observe that our source-only numbers are better than previously published results. Similar to previous studies (Gulrajani & Lopez-Paz, 2020), this can be attributed to improved design choices (e.g. data augmentation, hyperparameters) which we make consistent across all methods. While there is no consistent method that does the best across datasets, overall, FixMatch with RS and RW (our meta-algorithm) performs the best for vision datasets. For NLP datasets, source-only with RW (our meta-algorithm) performs the best overall. For tabular datasets, CDANN with RS and RW (our meta-algorithm) performs the best overall (Fig. 2).

**Existing DA methods when paired with our meta-algorithm significantly outperform other DA methods specifically proposed for relaxed label shift.** We observe that, with consistent experimental design across different methods, existing DA methods with RS and RW corrections often

improve over previously proposed methods specifically aimed to tackle relaxed label shift, i.e., IW-CDANN, IW-DANN, and SENTRY (Fig. 6). For severe target label marginal shifts, the performance of IW-DANN, IW-CDANN, and SENTRY often falls below that of the source-only model. Moreover, while the importance weighting (i.e., IW-CDANN and IW-DANN) improves over CDANN and DANN resp. (Fig. 1a, 4a and 5a), RS and RW corrections significantly outweigh those improvements (Fig. 6).

**BN-adapt and TENT with our meta-algorithm are simple and strong baselines.** For models with batch norm parameters, BN-adapt (and TENT) with RS and RW steps is a computationally efficient and strong baseline. We observe that while the performance of BN-adapt (and TENT) can drop substantially when the target label marginal shifts (i.e., $\alpha \in \{1.0, 0.5\}$ in Fig. 1(a)), RS and RW correction improves the performance often improving BN-adapt (and TENT) over all other DA methods when the shift in target label marginal is extreme (i.e., $\alpha = 0.5$ in Fig. 1(b)).

**Deep DA heuristics often improve target label marginal estimation on tabular and vision modalities.** Recall that we experiment with target label marginal estimation methods that leverage off-the-shelf classifiers to obtain an estimate. We observe that estimators leveraging DA classifiers tend to perform better than using source-only classifiers for tabular and vision datasets (vision in Fig. 3 and others in App. G). For NLP, we observe improved estimation with source-only model over DA classifiers. Correspondingly, as one might expect, better estimation yields greater accuracy improvements when applying RW correction. In particular, RW correction with DA methods improves over the source-only classifier for vision and tabular datasets and vice-versa for NLP datasets. (App. G). Moreover, for all modalities, we observe a trade-off between estimation error with the baseline method (i.e. binning target pseudolabels) and RLLS (or MLLS) method with severity in target marginal shift (Fig. 11).

**Early stopping criterion matters.** We observe a consistent $\approx 2\%$ and $\approx 8\%$ accuracy difference on vision and tabular datasets respectively with all methods (Fig. 12). On NLP datasets, while the early stopping criteria have $\approx 2\%$ accuracy difference when RW and RS corrections are not employed, the difference becomes negligible when these corrections are employed (Fig. 12). These results highlight that subsequent works should describe the early stopping criteria used within their evaluations.

**Data augmentation helps.** Corroborating findings from previous empirical studies in other settings (Gulrajani & Lopez-Paz, 2020; Sagawa et al., 2021), we observe that data augmentation techniques can improve the performance of a source-only model on vision datasets in relaxed label shift scenarios (refer to result on each dataset in App. I). Thus, whenever applicable, subsequent methods should use data augmentations.

## 5 CONCLUSION

Our work is the first large-scale study investigating methods under the relaxed label shift scenario. Relative to works operating strictly under the label shift assumption, RLSBENCH provides an opportunity for sensitivity analysis, allowing researchers to measure the robustness of their methods under various sorts of perturbations to the class-conditional distributions. Relative to the benchmark-driven deep domain adaptation literature, our work provides a comprehensive and standardized suite for evaluating under shifts in label distributions, bringing these benchmarks one step closer to exhibit the sort of diversity that we should expect to encounter when deploying models in the wild. On one hand, the consistent improvements observed from label shift adjustments are promising. At the same time, given the underspecified nature of the problem, practitioners must remain vigilant and take performance on any benchmark with a grain of salt, considering the various ways that it might (or might not) be representative of the sorts of situations that might arise in their application of interest.

In the future, we hope to extend RLSBENCH to datasets from real applications in consequential domains such as healthcare and self-driving, where both shifts in label prevalences and perturbations in class conditional distributions can be expected across locations and over time. We also hope to incorporate self-supervised methods that learn representations by training on a union of unlabeled data from source and target via proxy tasks like reconstruction (Gidaris et al., 2018; He et al., 2022) and contrastive learning (Caron et al., 2020; Chen et al., 2020). While re-weighting predictions using estimates of the target label distribution yields significant gains, the remaining gap between our results and oracle performance should motivate future work geared towards improved estimators. Also, we observe that the success of target label marginal estimation techniques depends on the nature of the shifts in $p(x|y)$. Mathematically characterizing the behavior of label shift estimation techniques when the label shift assumption is violated would be an important contribution.

## REPRODUCIBILITY STATEMENT

Our code with all the results is released on github: `https://github.com/ICLR2023Anon`.
We implement our RLSBENCH library in PyTorch (Paszke et al., 2017) and provide an infrastructure
to run all the experiments to generate corresponding results. We have stored all models and logged
all hyperparameters and seeds to facilitate reproducibility. In our appendices, we provide additional
details on datasets and experiments. In App. D, we describe dataset information and in App. L, we
describe hyperparameter details.

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

APPENDIX

# A DESCRIPTION OF PLOTS

For each plot in Fig. 1, we obtain all the distribution shift pairs with a specific alpha (i.e., the value on the x-axis). Then for each distribution shift pair (with a specific alpha value), we obtain *relative performance* by subtracting the performance of a source-only model trained on the source dataset of that distribution shift pair from the performance of the model trained on that distribution shift pair with the DA algorithm of interest. Thus for each alpha and each DA method, we obtain 112 relative performance values. We draw the box plot and the mean of these relative performance values.

For (similar-looking) plots, we use the same technique throughout the paper. The only thing that changes is the group of points over which aggregation is performed.

# B TABULAR AND NLP RESULTS OMITTED FROM THE MAIN PAPER

## B.1 TABULAR DATASETS

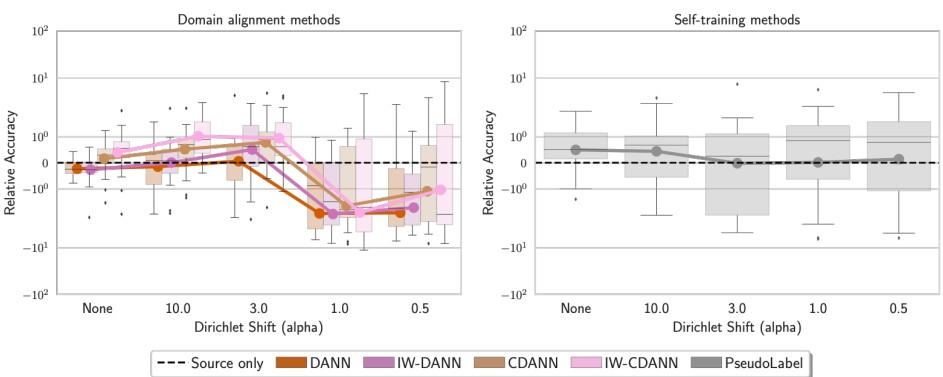

(a) Performance of DA methods relative to source-only training with increasing target label marginal shift

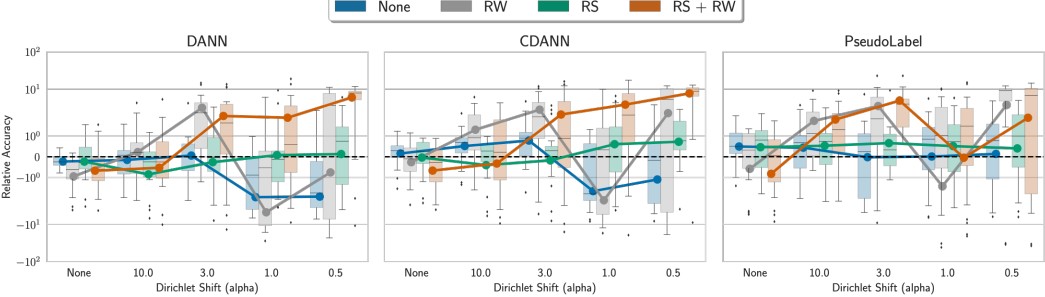

(b) Relative performance of DA methods when paired with our meta-algorithm (RS and RW corrections)

Figure 4: *Performance of different DA methods relative to a source-only model across all distribution shift pairs in tabular datasets grouped by shift severity in label marginal.* For each distribution shift pair and DA method, we plot the relative accuracy of the model trained with that DA method by subtracting the accuracy of the source-only model. Hence, the black dotted line at 0 captures the performance of the source-only model. Smaller the Dirichlet shift parameter, the more severe is the shift in target class proportion. **(a)** Shifts with $\alpha = \{\text{NONE}, 10.0, 3.0\}$ have little to no impact on different DA methods whereas the performance of all DA methods degrades when $\alpha \in \{1.0, 0.5\}$ often falling below the performance of a source-only classifier. **(b)** RS and RW (in our meta-algorithm) together significantly improve aggregate performance over no correction for all DA methods. While RS consistently helps (over no correction) across different label marginal shift severities, RW hurts slightly when shift severity is small. However, for severe shifts ($\alpha \in \{3.0, 1.0, 0.5\}$) RW significantly improves performance for all the methods. Results with all methods on individual datasets in App. I.

## B.2 NLP DATASETS

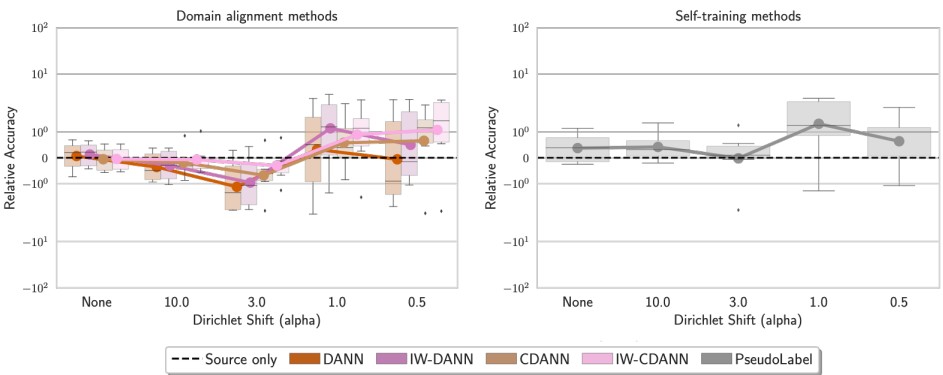

(a) Performance of DA methods relative to source-only training with increasing target label marginal shift

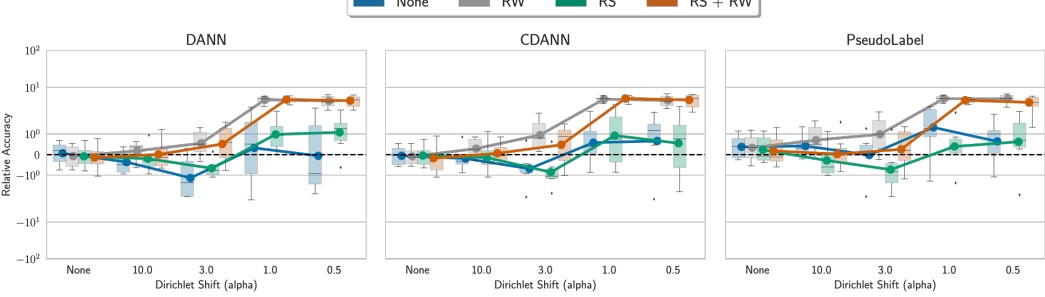

(b) Relative performance of DA methods when paired with our meta-algorithm (RS and RW corrections)

Figure 5: *Performance of different DA methods relative to a source-only model across all distribution shift pairs in NLP datasets grouped by shift severity in label marginal.* For each distribution shift pair and DA method, we plot the relative accuracy of the model trained with that DA method by subtracting the accuracy of the source-only model. Hence, the black dotted line at 0 captures the performance of the source-only model. Smaller the Dirichlet shift parameter, the more severe is the shift in target class proportion. **(a)** Performance of DANN and IW-DANN methods degrades with increasing severity of target label marginal shift often falling below the performance of a source-only classifier (except for Noisy Student). Performance of PsuedoLabel, CDANN, and IW-CDANN show less susceptibility to increasing severity in target marginal shift. **(b)** RS and RW (in our meta-algorithm) together significantly improve aggregate performance over no correction for all DA methods. While RS consistently helps (over no correction) across different label marginal shift severities, RW hurts slightly for BN-adapt, TENT, and NoisyStudent when shift severity is small. However, for severe shifts ($\alpha \in \{3.0, 1.0, 0.5\}$) RW significantly improves performance for all the methods. Detailed results with all methods on individual datasets in App. I.

## C COMPARISON BETWEEN IW-CDANN, IW-DANN, AND SENTRY WITH EXISTING DA METHODS PAIRED WITH OUR META-ALGORITHM

Fig. 6 shows the relevant comparison.

**Note.** On Officehome dataset, we observe a slight discrepancy between SENTRY results with our runs and numbers originally reported in the paper (Prabhu et al., 2021). We find that this is due to differences in batch size used in original work versus in our runs (which we kept the same for all the algorithms). In App. M, we report SENTRY results with the updated batch size. With the new batch size, we reconcile SENTRY results but also observe a significant improvement in FixMatch results. We refer reader to App. M for a more detailed discussion.

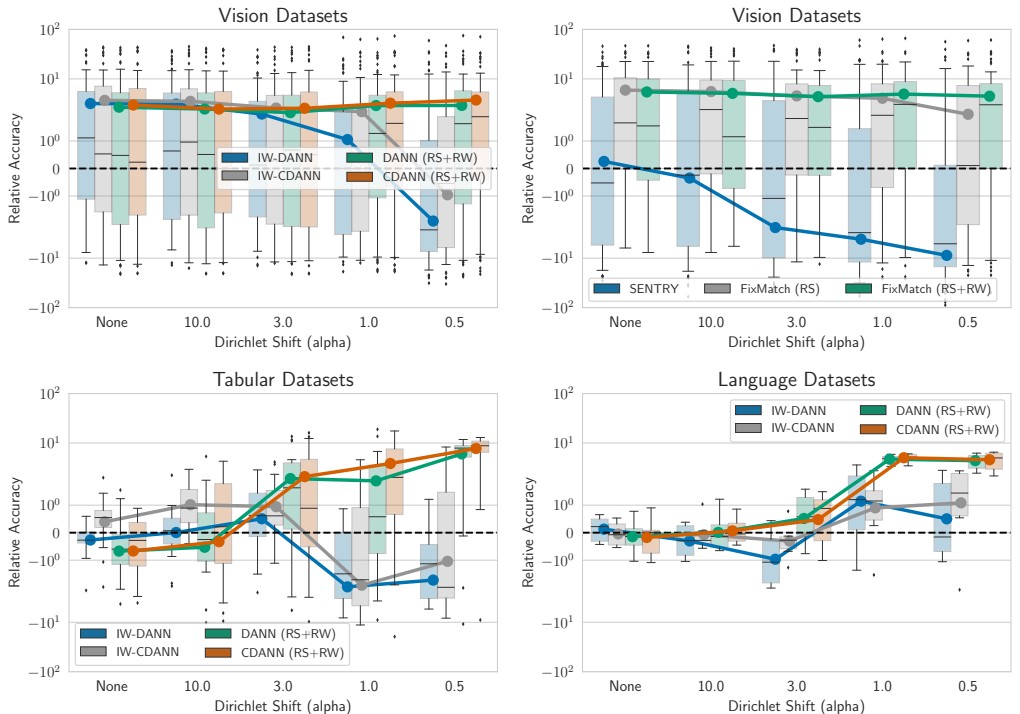

Figure 6: *Comparison of existing DA methods paired with our RS and RW correction and DA methods specifically proposed for relaxed label shift problems.* Across vision and tabular datasets, we observe the susceptibility of IW-DAN, IW-CDAN, and SENTRY with increasing severity of target label marginal shifts. In particular, for severe target label marginal shifts, the performance of IW-DAN, IW-CDAN, and SENTRY often falls below that of the source-only model. However, existing DA techniques when paired with RS + RW correction significantly improve over the source-only model. For NLP, datasets we observe similar behavior but with relatively less intensity.

## D  DATASET DETAILS

In this section, we provide additional details about the datasets used in our benchmark study.

- **CIFAR10**  We use the original CIFAR10 dataset (Krizhevsky & Hinton, 2009) as the source dataset. For target domains, we consider (i) synthetic shifts (CIFAR10-C) due to common corruptions (Hendrycks & Dietterich, 2019); and (ii) natural distribution shift, i.e., CIFAR10v2 (Recht et al., 2018; Torralba et al., 2008) due to differences in data collection strategy. We randomly sample 3 set of CIFAR-10-C datasets. Overall, we obtain 5 datasets (i.e., CIFAR10v1, CIFAR10v2, CIFAR10C-Frost (severity 4), CIFAR10C-Pixelate (severity 5), CIFAR10-C Saturate (severity 5)).

- **CIFAR100**  Similar to CIFAR10, we use the original CIFAR100 set as the source dataset. For target domains we consider synthetic shifts (CIFAR100-C) due to common corruptions. We sample 4 CIFAR100-C datasets, overall obtaining 5 domains (i.e., CIFAR100, CIFAR100C-Fog (severity 4), CIFAR100C-Motion Blur (severity 2), CIFAR100C-Contrast (severity 4), CIFAR100C-spatter (severity 2) ).

- **FMoW**  In order to consider distribution shifts faced in the wild, we consider FMoW-WILDs (Koh et al., 2021; Christie et al., 2018) from WILDS benchmark, which contains satellite images taken in different geographical regions and at different times. We use the original train as source and OOD val and OOD test splits as target domains as they are collected over different time-period. Overall, we obtain 3 different domains.

- **Camelyon17**  Similar to FMoW, we consider tumor identification dataset from the wilds benchmark (Bandi et al., 2018). We use the default train as source and OOD val and OOD test splits as target domains as they are collected across different hospitals. Overall, we obtain 3 different domains.

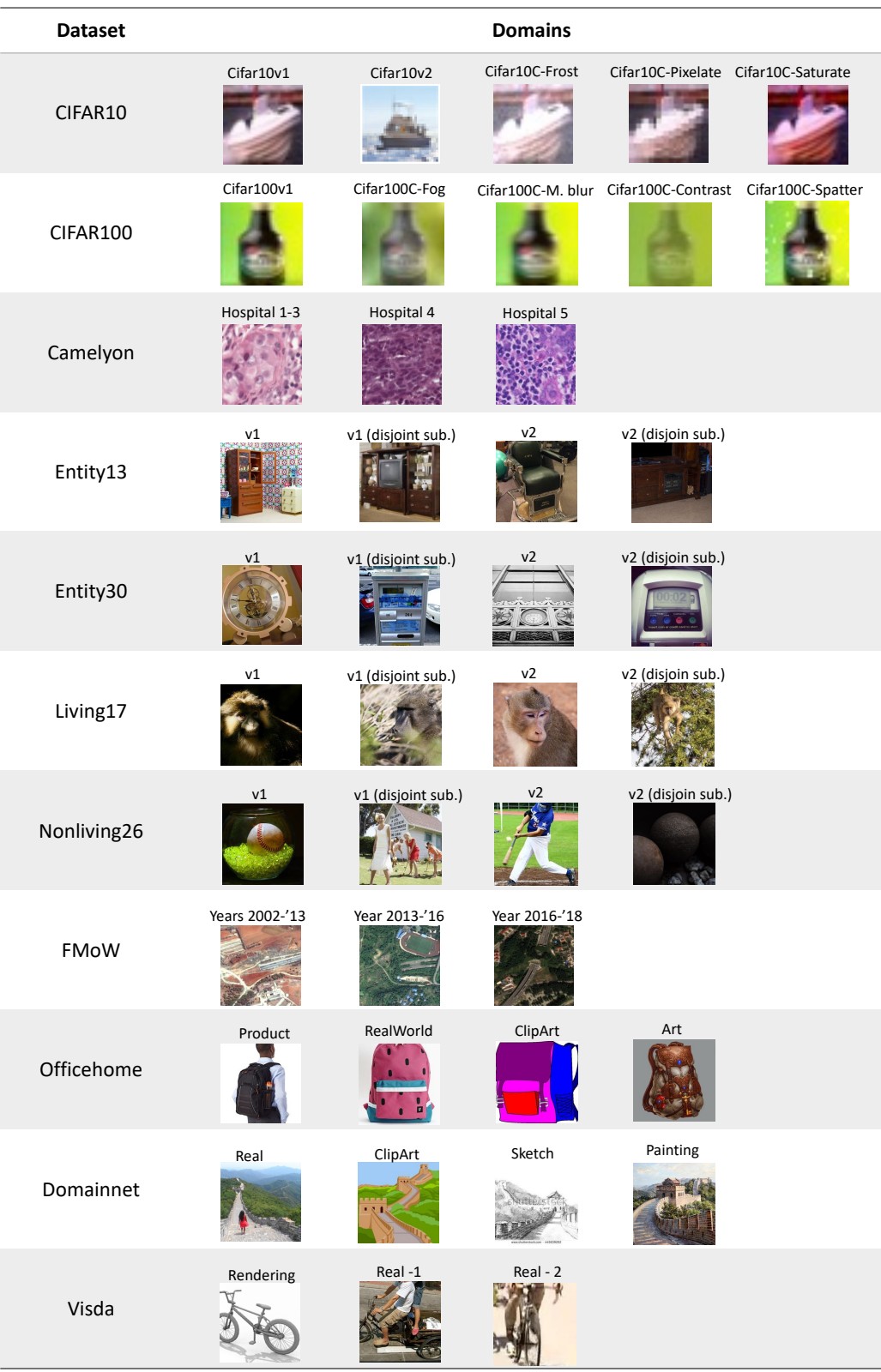

Figure 7: Examples from all the domains in each vision dataset.

| Dataset | Source | Target |
|---|---|---|
| CIFAR10 | CIFAR10v1 | CIFAR10v1, CIFAR10v2, CIFAR10C-Frost (severity 4), CIFAR10C-Pixelate (severity 5), CIFAR10-C Saturate (severity 5) |
| CIFAR100 | CIFAR100 | CIFAR100, CIFAR100C-Fog (severity 4), CIFAR100C-Motion Blur (severity 2), CIFAR100C-Contrast (severity 4), CIFAR100C-spatter (severity 2) |
| Camelyon | Camelyon (Hospital 1–3) | Camelyon (Hospital 1–3), Camelyon (Hospital 4), Camelyon (Hospital 5) |
| FMoW | FMoW (2002–'13) | FMoW (2002–'13), FMoW (2013–'16), FMoW (2016–'18) |
| Entity13 | Entity13 (ImageNetv1 sub-population 1) | Entity13 (ImageNetv1 sub-population 1), Entity13 (ImageNetv1 sub-population 2), Entity13 (ImageNetv2 sub-population 1), Entity13 (ImageNetv2 sub-population 2) |
| Entity30 | Entity30 (ImageNetv1 sub-population 1) | Entity30 (ImageNetv1 sub-population 1), Entity30 (ImageNetv1 sub-population 2), Entity30 (ImageNetv2 sub-population 1), Entity30 (ImageNetv2 sub-population 2) |
| Living17 | Living17 (ImageNetv1 sub-population 1) | Living17 (ImageNetv1 sub-population 1), Living17 (ImageNetv1 sub-population 2), Living17 (ImageNetv2 sub-population 1), Living17 (ImageNetv2 sub-population 2) |
| Nonliving26 | Nonliving26 (ImageNetv1 sub-population 1) | Nonliving26 (ImageNetv1 sub-population 1), Nonliving26 (ImageNetv1 sub-population 2), Nonliving26 (ImageNetv2 sub-population 1), Nonliving26 (ImageNetv2 sub-population 2) |
| Officehome | Product | Product, Art, ClipArt, Real |
| DomainNet | Real | Real, Painting, Sketch, ClipArt |
| Visda | Synthetic (originally referred to as train) | Synthetic, Real-1 (originally referred to as val), Real-2 (originally referred to as test) |
| Civilcomments | Train | Train, Val and Test (all formed by disjoint partitions of online articles) |
| Mimic Readmissions | Mimic Readmissions (year: 2008) | Mimic Readmissions (year: 2008), Mimic Readmissions (year: 2009), Mimic Readmissions (year: 2010), Mimic Readmissions (year: 2011), Mimic Readmissions (year: 2012), Mimic Readmissions (year: 2013) |
| Retiring Adults | Retiring Adults (year: 2014 states: ['MD', 'NJ', 'MA']) | Retiring Adults (year: 2015; states: ['MD', 'NJ', 'MA']), Retiring Adults (year: 2016; states: ['MD', 'NJ', 'MA']), Retiring Adults (year: 2017; states: ['MD', 'NJ', 'MA']), Retiring Adults (year: 2018; states: ['MD', 'NJ', 'MA']) |

Table 1: Details of the datasets considered in our RLSBENCH.

- **BREEDs** We also consider BREEDs benchmark (Santurkar et al., 2021) in our setup to assess robustness to subpopulation shifts. BREEDs leverage class hierarchy in ImageNet to re-purpose original classes to be the subpopulations and defines a classification task on superclasses. We consider distribution shift due to subpopulation shift which is induced by directly making the subpopulations present in the training and test distributions disjoint. BREEDs benchmark contains 4 datasets **Entity-13**, **Entity-30**, **Living-17**, and **Non-living-26**, each focusing on different subtrees and levels in the hierarchy. We also consider natural shifts due to differences in the data collection process of ImageNet (Russakovsky et al., 2015), e.g, ImageNetv2 (Recht et al., 2019) and a combination of both. Overall, for each of the 4 BREEDs datasets (i.e., Entity-13, Entity-30, Living-17, and Non-living-26), we obtain four different domains. We refer to them as follows: BREEDsv1 sub-population 1 (sampled from ImageNetv1), BREEDsv1 sub-population 2 (sampled from ImageNetv1), BREEDsv2 sub-population 1 (sampled from ImageNetv2), BREEDsv2 sub-population 2 (sampled from ImageNetv2). For each BREEDs dataset, we use BREEDsv1 sub-population A as source and the other three as target domains.

- **OfficeHome** We use four domains (art, clipart, product and real) from OfficeHome dataset (Venkateswara et al., 2017). We use the product domain as source and the other domains as target.

- **DomainNet** We use four domains (clipart, painting, real, sketch) from the Domainnet dataset (Peng et al., 2019). We use real domain as the source and the other domains as target.

- **Visda** We use three domains (train, val and test) from the Visda dataset (Peng et al., 2018). While 'train' domain contains synthetic renditions of the objects, 'val' and 'test' domains contain real world images. To avoid confusing, the domain names with their roles as splits, we rename them as 'synthetic', 'Real-1' and 'Real-2'. We use the synthetic (original train set) as the source domain and use the other domains as target.

- **Civilcomments** (Borkan et al., 2019) from the wilds benchmark which includes three domains: train, OOD val, and OOD test, for toxicity detection with domains corresponding to different demographic subpopulations. The dataset has subpopulation shift across different demographic groups as the dataset in each domain is collected from a different partition of online articles.

- **Retiring Adults** (Ding et al., 2021) where we consider the ACSIncome prediction task with various domains representing different states and time-period; We randomly select three states and consider dataset due to shifting time across those states. Details about precise time-periods and states are in Table 1.

- **Mimic Readmission** (Johnson et al., 2020; PhysioBank, 2000) where the task is to predict readmission risk with various domains representing data from different time-period. Details about precise time-periods are in Table 1.

We provide scripts to setup these datasets with single command in our code. To investigate the performance of different methods under the stricter label shift setting, we also include a hold-out partition of source domain in the set of target domains. For these distribution shift pairs where source and target domains are i.i.d. partitions, we obtain the stricter label shift problem. We summarize the information about source and target domains in Table 1.

**Train-test splits** We partition each source and target dataset into $80\%$ and $20\%$ i.i.d. splits. We use $80\%$ splits for training and $20\%$ splits for evaluation (or validation). We throw away labels for the $80\%$ target split and only use labels in the $20\%$ target split for final evaluation. The rationale behind splitting the target data is to use a completely unseen batch of data for evaluation. This avoids evaluating on examples where a model potentially could have overfit. over-fitting to unlabeled examples for evaluation. In practice, if the aim is to make predictions on all the target data (i.e., transduction), we can simply use the (full) target set for training and evaluation.

## E METHODS TO ESTIMATE TARGET MARGINAL UNDER THE STRICTER LABEL SHIFT ASSUMPTION

In this section, we describe the methods proposed to estimate the target label marginal under the stricter label shift assumption. Recall that under the label shift assumption, $p_s(y)$ can differ from $p_t(y)$ but the class conditional stays the same, i.e., $p_t(x|y) = p_s(x|y)$. We focus our discussion on recent methods that leverage off-the-shelf classifier to yield consistent estimates under mild assumptions (Lipton et al., 2018; Azizzadenesheli et al., 2019; Alexandari et al., 2021; Garg et al., 2020). For simplicity, we assume we possess labeled source data $\{(x_1, y_1), (x_2, y_2), \ldots, (x_n, y_n)\}$ and unlabeled target data $\{x_{n+1}, x_{n+2}, \ldots, x_{n+m}\}$.

**RLLS** First, we discuss *Regularized Learning under Label Shift* (RLLS) (Azizzadenesheli et al., 2019) (a variant of *Black Box Shift Estimation* (BBSE, Lipton et al. (2018))): moment-matching based estimators that leverage (possibly biased, uncalibrated, or inaccurate) predictions to estimate the shift. RLLS solves the following optimization problem to estimate the importance weights $w_t(y) = \frac{p_t(y)}{p_s(y)}$ as:

$$\widehat{w}_t^{\text{RLLS}} = \arg\min_{w \in \mathcal{W}} \left\| \widehat{C}_f w - \widehat{\mu}_f \right\|_2 + \lambda_{\text{RLLS}} \left\| w - 1 \right\|_2 . \tag{1}$$

where $\mathcal{W} = \{ w \in \mathbb{R}^d \,|\, \sum_y w(y) p_s(y) = 1 \text{ and } \forall y \in \mathcal{Y} \quad w(y) > 0 \}$. $\widehat{C}_f$ is empirical confusion matrix of the classifier $f$ on source data and $\widetilde{\mu}_f$ is the empirical average of predictions of the classifier $f$ on unlabeled target data. With labeled source data data, the empirical confusion matrix can be

computed as:

$$[\widehat{C}_f]_{i,j} = \frac{1}{n} \sum_{k=1}^{n} f_i(x_k) \cdot \mathbb{I}\left[y_k = j\right] .$$

To estimate target label marginal, we can multiple the estimated importance weights with the source label marginal (we can estimate source label marginal simply from labeled source data).

In our relaxed label shift problem, we use validation source data to compute the confusion matrix and use hold portion of target unlabeled data to compute $\mu_f$. Unless specified otherwise, we use RLLS to estimate the target label marginal throughout the paper. We choose $\lambda_{\mathrm{RLLS}}$ as suggested in the original paper (Azizzadenesheli et al., 2019).

**MLLS**  Next, we discuss Maximum Likelihood Label Shift (MLLS) (Saerens et al., 2002; Alexandari et al., 2021): an Expectation Maximization (EM) algorithm that maximize the likelihood of observed unlabeled target data to estimate target label marginal assuming access to a classifier that outputs the source calibrated probabilities. In particular, MLLS uses the following objective:

$$\widehat{w}_t^{\mathrm{MLLS}} = \arg\min_{w \in \mathcal{W}} \frac{1}{m} \sum_{i=1} \log(w^T f(x_{i+n})) , \tag{2}$$

where $f$ is the classifier trained on source and $\mathcal{W}$ is the same constrained set defined above. We can again estimate the target label marginal by simply multiplying the estimated importance weights with the source label marginal.

**Baseline estimator**  Given a classifier $f$, we can estimate the target label marginal as simply the average of the classifier output on unlabeled target data, i.e.,

$$\widehat{p}_t^{\mathrm{baseline}} = \frac{1}{m} \sum_{i=1} f(x_{i+n}) . \tag{3}$$

Note that all of the methods discussed before leverage an off-the-shelf classifier $f$. Hence, we experiment with classifiers obtained with various deep domain adaptation heuristics to estimate the target label marginal.

Having obtained an estimate of target label marginal, we can simply re-weight the classifier with $\widehat{p}_t$ as $f'_j = \frac{\widehat{p}_t(y=j) \cdot f_j}{\sum_k \widehat{p}_t(y=k) \cdot f_k}$ for all $j \in \mathcal{Y}$. Note that, if we train $f$ on a non-uniform source class-balance (and without re-balancing as in Step 1 of Algorithm 1), then we can re-weight the classifier with importance-weights $\widehat{w}_t$ as $f'_j = \frac{\widehat{w}_t(y=j) \cdot f_j}{\sum_k \widehat{w}_t(y=k) \cdot f_k}$ for all $j \in \mathcal{Y}$.

## F  THEORETICAL DEFINITION FOR RELAXED LABEL SHIFT

Domain adaptation problems are, in general, ill-posed (Ben-David et al., 2010b). Several attempts have been made to investigate additional assumptions that render the problem well-posed. One such example includes the label-shift setting, where $p(x|y)$ does not change but that $p(y)$ can. Under label shift, two challenges arise: (i) estimate the target label marginal $p_t(y)$; and (ii) train a classifier $f$ to maximize the performance on the target domain. However, these assumptions are typically, to some degree, violated in practice. This paper aims to relax this assumption and focuses on *relaxed label shift* setting. In particular, we assume that the label distribution can shift from source to target arbitrarily but that $p(x|y)$ varies between source and target in some comparatively restrictive way (e.g., shifts arising naturally in the real world like ImageNet (Russakovsky et al., 2015) to ImageNetV2 (Recht et al., 2019)).

Mathematically, we assume a divergence-based restriction on $p(x|y)$, i.e., for some small $\epsilon > 0$ and distributional distance $\mathcal{D}$, we have $\max_y \mathcal{D}(p_t(x|y), p_t(x|y)) \leqslant \epsilon$ but allowing an arbitrary shift in the label marginal $p(y)$. Previous works have defined these constraints in different ways (Wu et al., 2019; Tachet des Combes et al., 2020; Kumar et al., 2020).

In particular, we can use Wasserstein-infinity distance to define our constraint. First, we define Wasserstein given probability measures $p, q$ on $\mathcal{X}$:

$$W_\infty(p, q) = \inf\{ \sup_{x \in \mathbb{R}^d} \|f(x) - x\|_2 : f : \mathbb{R}^d \to \mathbb{R}^d, f_\# p = q \},$$

where # denotes the push forward of a measure, i.e., for every set $S \subseteq \mathbb{R}^d, p(S) = p(f^{-1}(S))$. Intuitively, $W_\infty$ moves points from the distribution $p$ to $q$ by distance at most $\epsilon$ to match the distributions. Hence, our $D := \max_y W_\infty(p_s(x|y), p_t(x|y)) \leqslant \epsilon$. Similarly, we can define our distribution constraint in KL or TV distances. We can define our constraint in a representation space $\mathcal{Z}$ obtained by projection inputs $x \in \mathcal{X}$ with a function $h : \mathcal{X} \to \mathcal{Z}$. Intuitively, we want to define the distribution distance with some $h$ that captures all the required information for predicting the label of interest but satisfies a small distributional divergence in the projected space. However, in practice, it's hard to empirically verify these distribution distances for small enough $\epsilon$ with finite samples. Moreover, we lack a rigorous characterization of the sense in which those shifts arise in popular DA benchmarks, and since, the focus of our work is on the empirical evaluation with real-world datasets, we leave a formal investigation for future work. .

# G    TARGET MARGINAL ESTIMATION AND ITS EFFECT ON ACCURACY

## G.1    VISION DATASETS

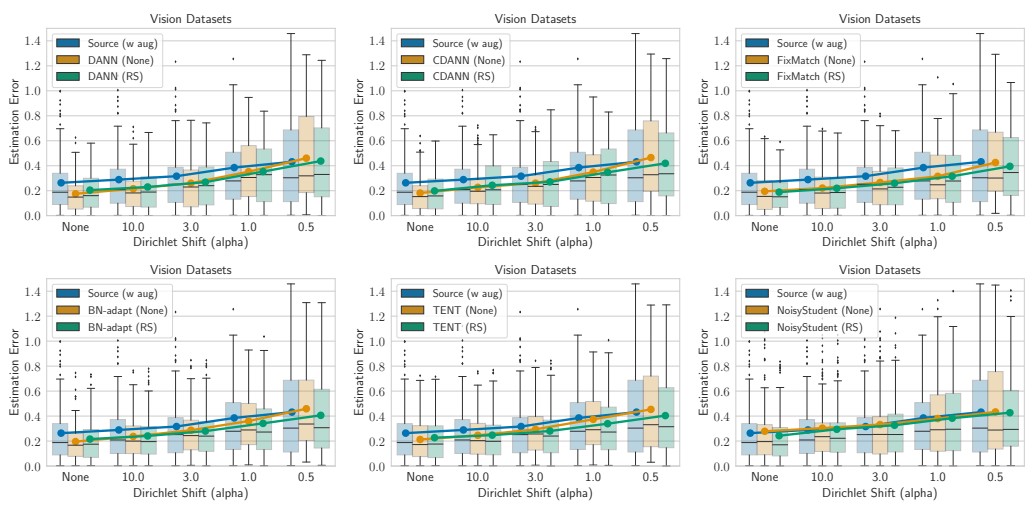

(a) Target label marginal estimation ($\ell_1$) error with RLLS and classifiers obtained with different DA methods

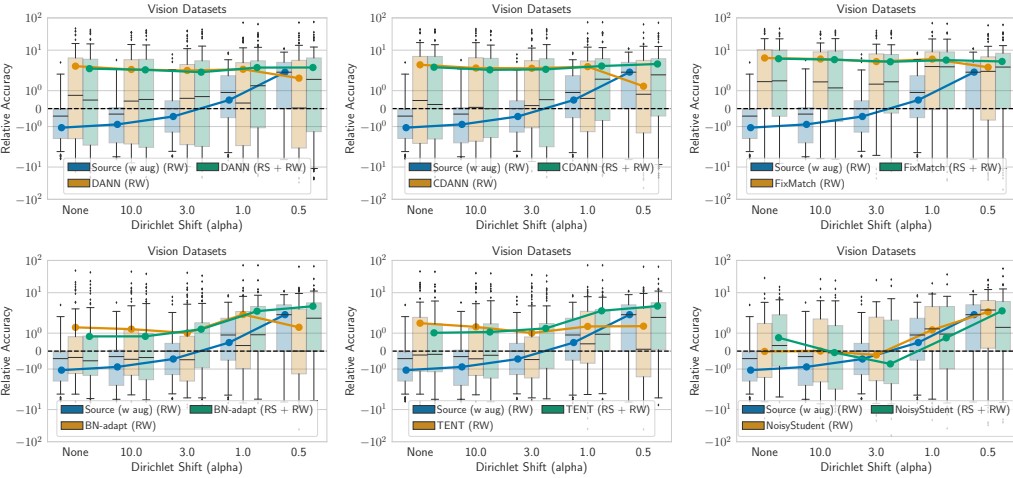

(b) Relative performance of DA methods when paired with RW corrections

Figure 8: *Target label marginal estimation ($\ell_1$) error and relative performance with RLLS and classifiers obtained with different DA methods.* Across all shift severities (except for $\alpha = 0.5$) in vision datasets, RLLS with classifiers obtained with DA methods improves over RLLS with a source-only classifier. Correspondingly, we see significantly improved performance with post-hoc RW correction applied to classifiers trained with DA methods as compared to when applied to source-only models.

## G.2 TABULAR DATASETS

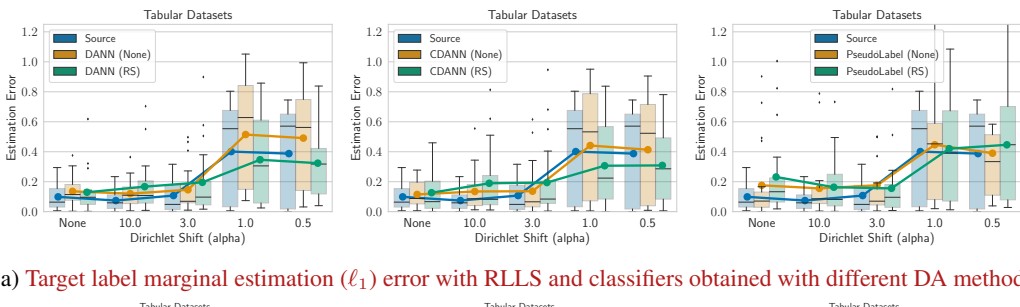

(a) Target label marginal estimation ($\ell_1$) error with RLLS and classifiers obtained with different DA methods

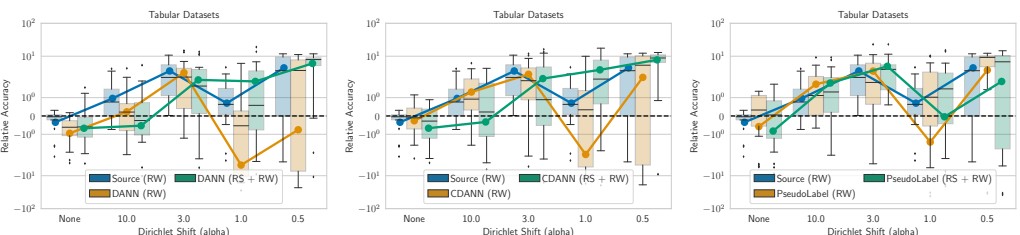

(b) Relative performance of DA methods when paired with RW corrections

Figure 9: *Target label marginal estimation ($\ell_1$) error and relative performance with RLLS and classifiers obtained with different DA methods.* For tabular datasets, RLLS with classifiers obtained with DA methods improves over RLLS with a source-only classifier for severe target label marginal shifts. Correspondingly for severe target label marginal shifts, we see improved performance with post-hoc RW correction applied to classifiers trained with DA methods as compared to when applied to source-only models.

## G.3 NLP DATASETS

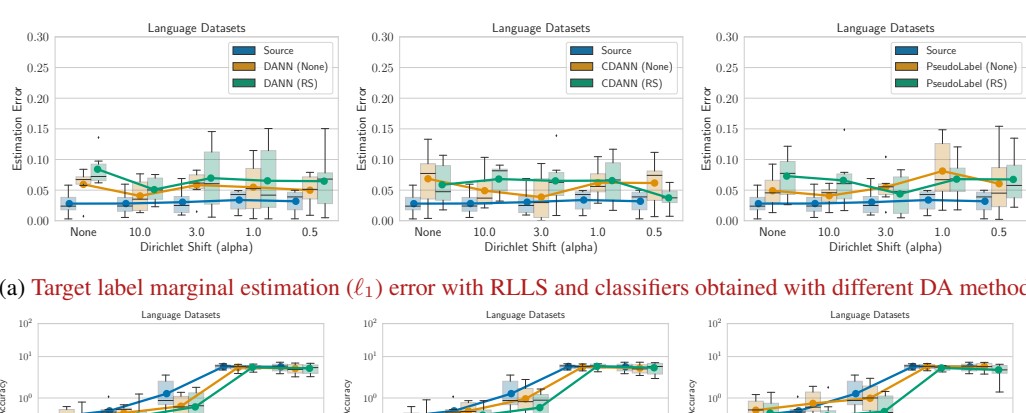

(a) Target label marginal estimation ($\ell_1$) error with RLLS and classifiers obtained with different DA methods

(b) Relative performance of DA methods when paired with RW corrections

Figure 10: *Target label marginal estimation ($\ell_1$) error and relative performance with RLLS and classifiers obtained with different DA methods.* For NLP datasets, RLLS with source-only classifiers performs better than RLLS with classifiers obtained with DA methods. Correspondingly, we see improved performance with post-hoc RW correction applied to source-only models over classifiers trained with DA methods.

### G.4 COMPARISON OF DIFFERENT TARGET LABEL MARGINAL ESTIMATION METHODS

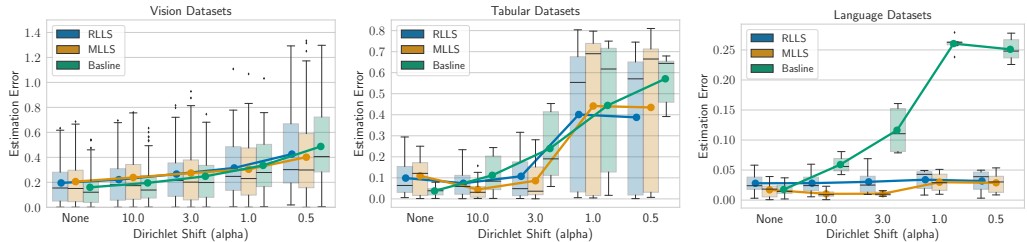

Figure 11: *Comparison of different target label marginal estimation methods.* We plot estimation errors with different methods with the source-only classifier. For all modalities, we observe a trade-off between estimation error with the baseline method and RLLS (or MLLS) method with severity in target marginal shift.

## H RESULTS WITH ORACLE EARLY STOPPING CRITERION

In this section, we report results with oracle early stopping criterion. On vision and tabular datasets, we observe differences in performance when using target performance versus source hold-out performance for model selection. This highlights a more nuanced behavior than the accuracy-on-the-line phenomena (Miller et al., 2021; Recht et al., 2019). We hope to study this contrasting behavior in more detail in future work.

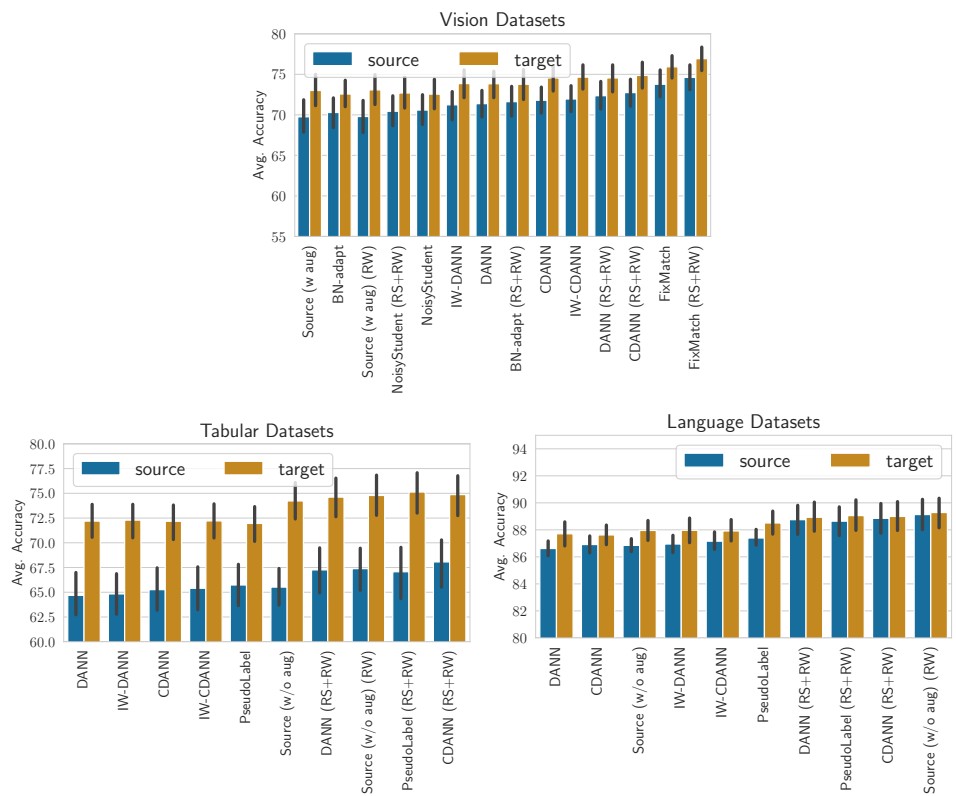

Figure 12: *Average accuracy of different DA methods aggregated across all distribution pairs in each modality.* We compare the performance with early stopping point obtained with source validation performance and target validation performance.

# I RESULTS ON INDIVIDUAL DATASETS

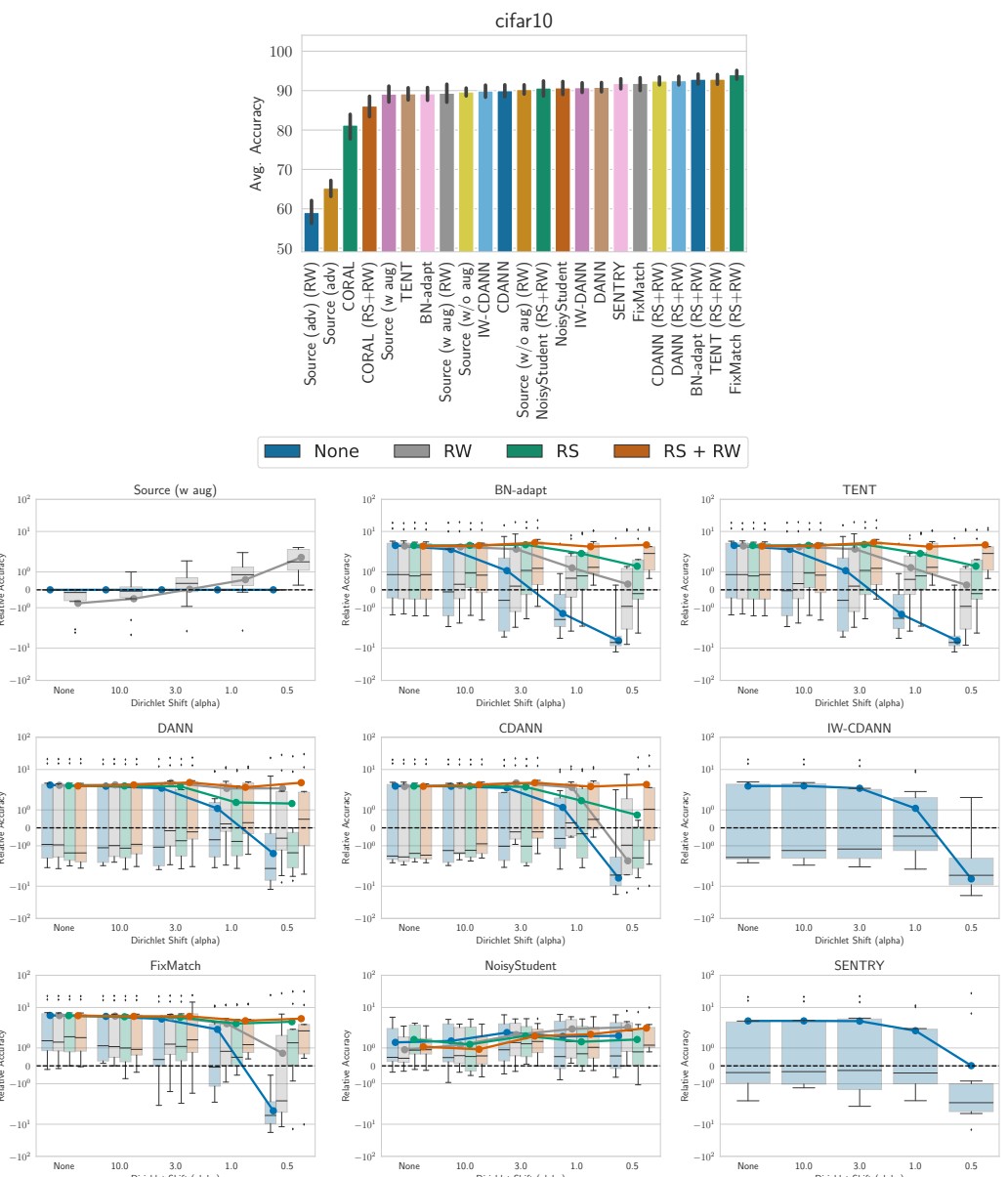

Figure 13: CIFAR10. Relative performance and accuracy plots for different DA algorithms across various shift pairs in CIFAR10.

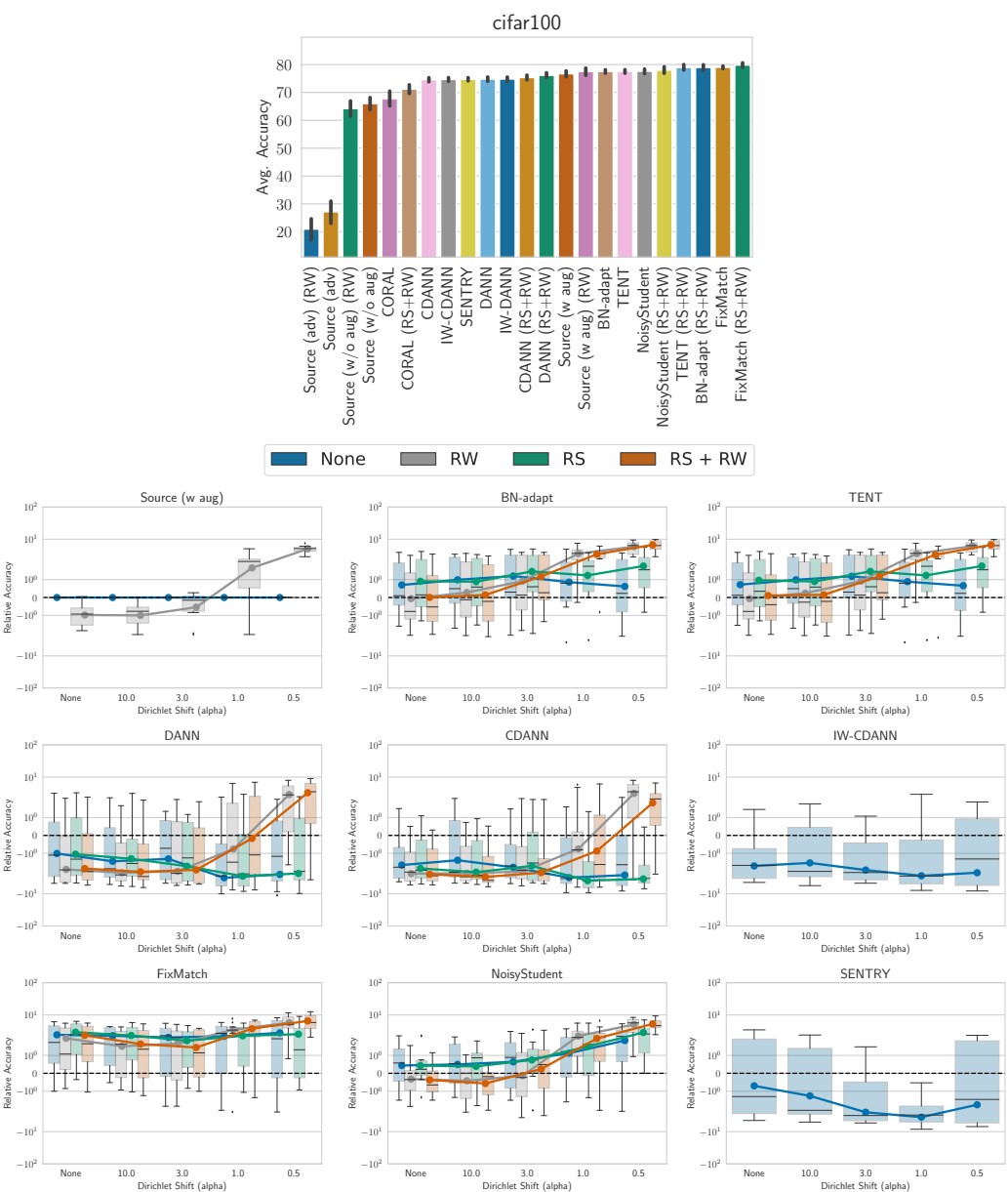

Figure 14: CIFAR100. Relative performance and accuracy plots for different DA algorithms across various shift pairs in CIFAR100.

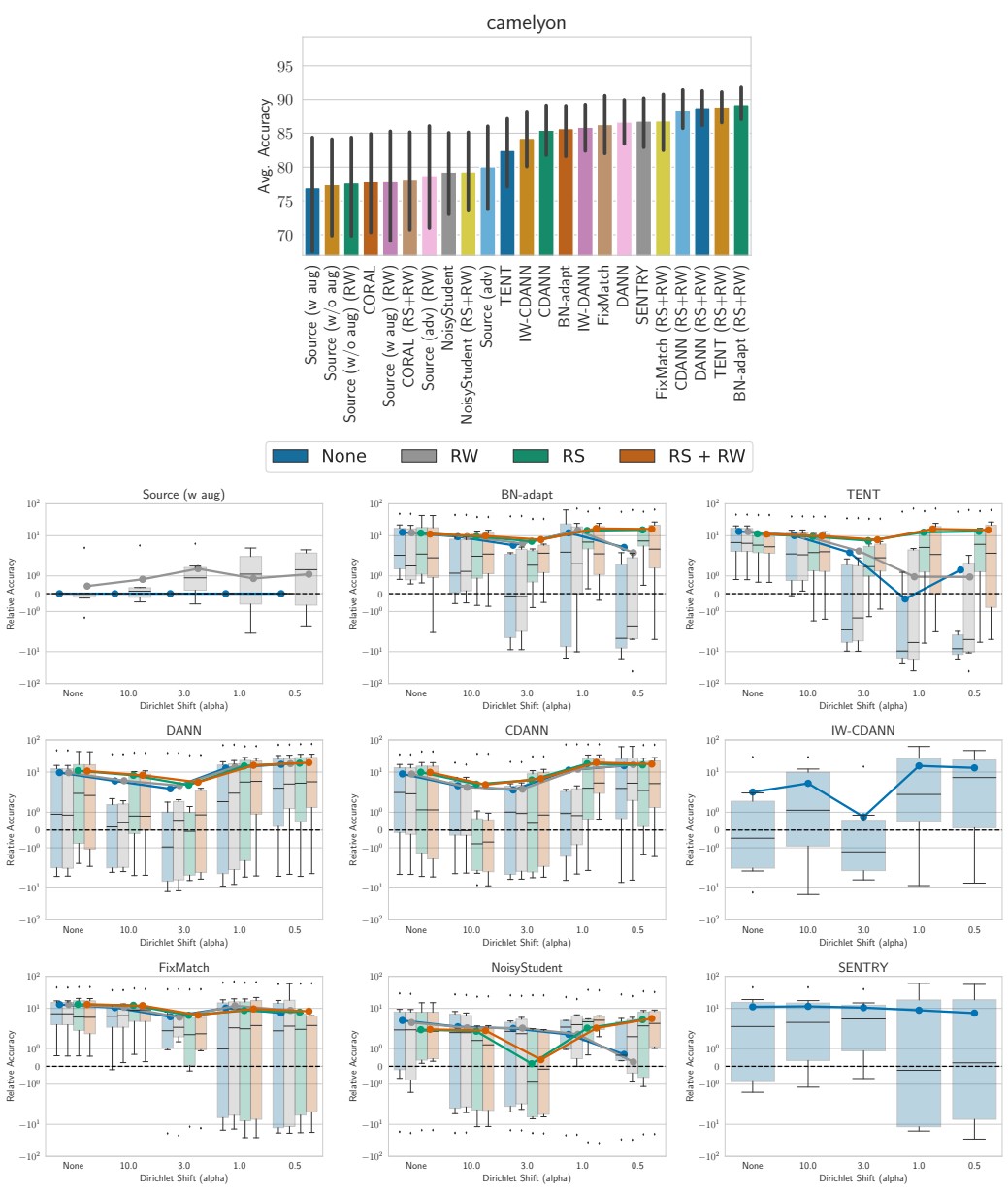

Figure 15: Camelyon. Relative performance and accuracy plots for different DA algorithms across various shift pairs in Camelyon.

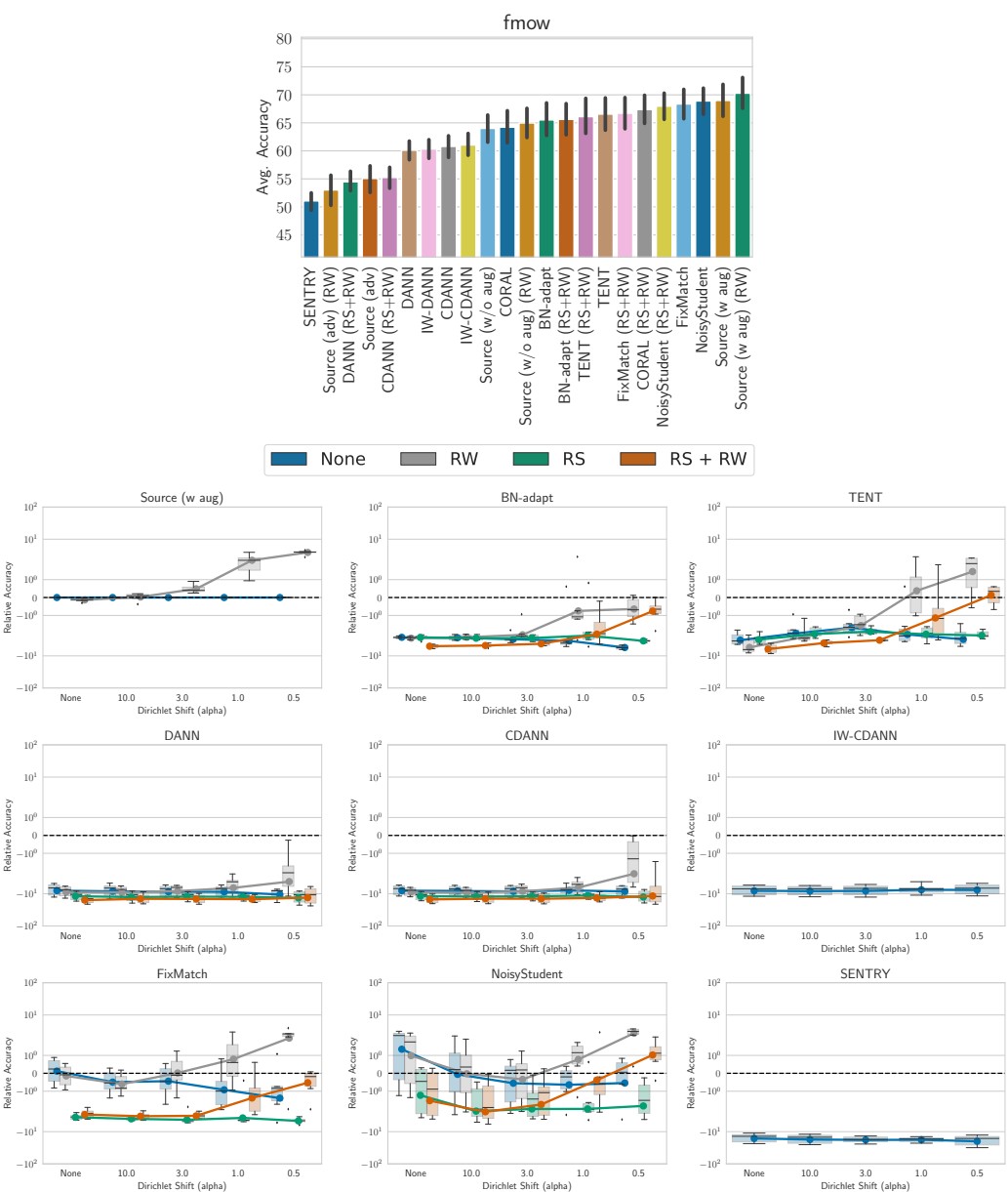

Figure 16: FMoW. Relative performance and accuracy plots for different DA algorithms across various shift pairs in FMoW.

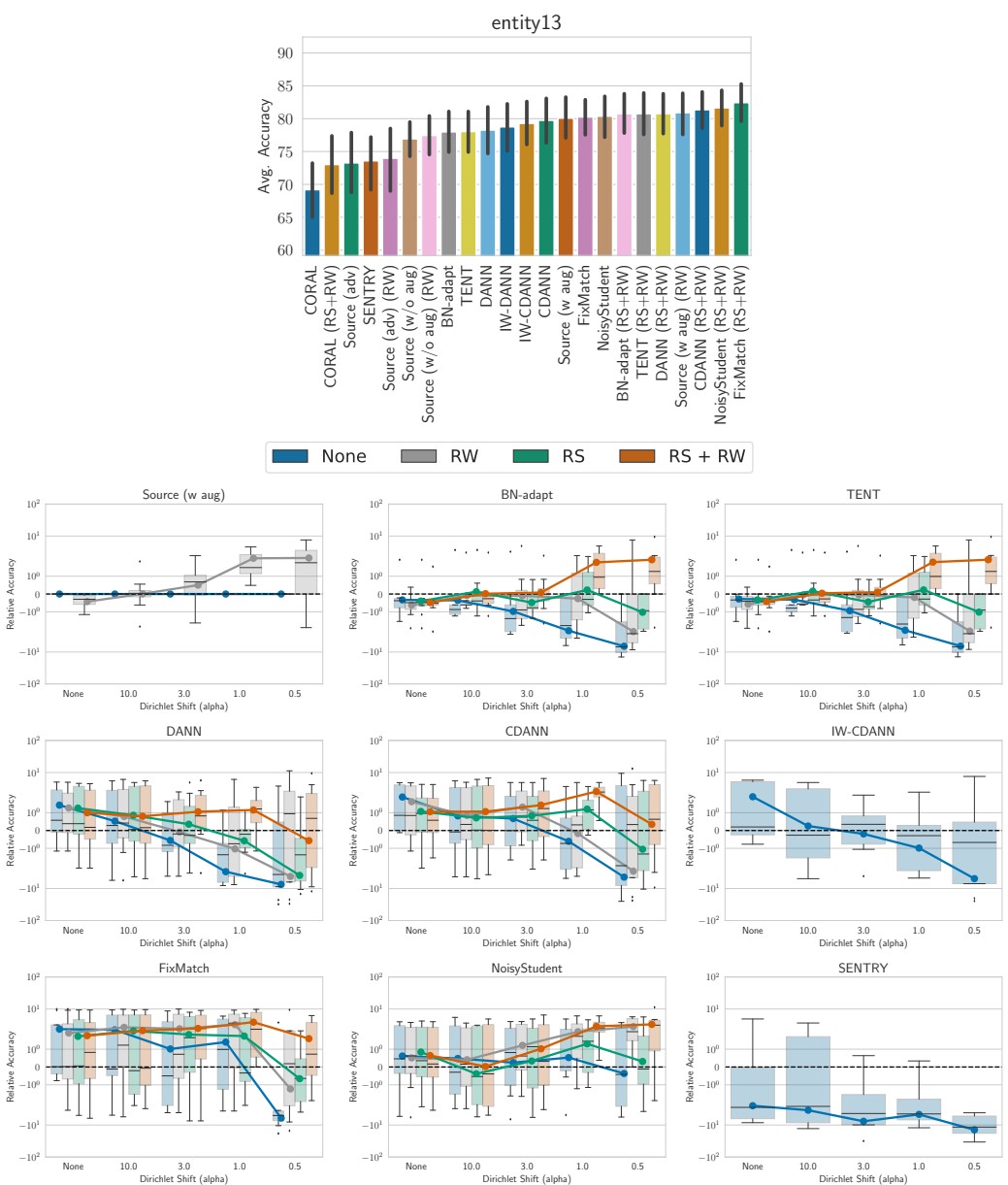

Figure 17: Entity13. Relative performance and accuracy plots for different DA algorithms across various shift pairs in Entity13.

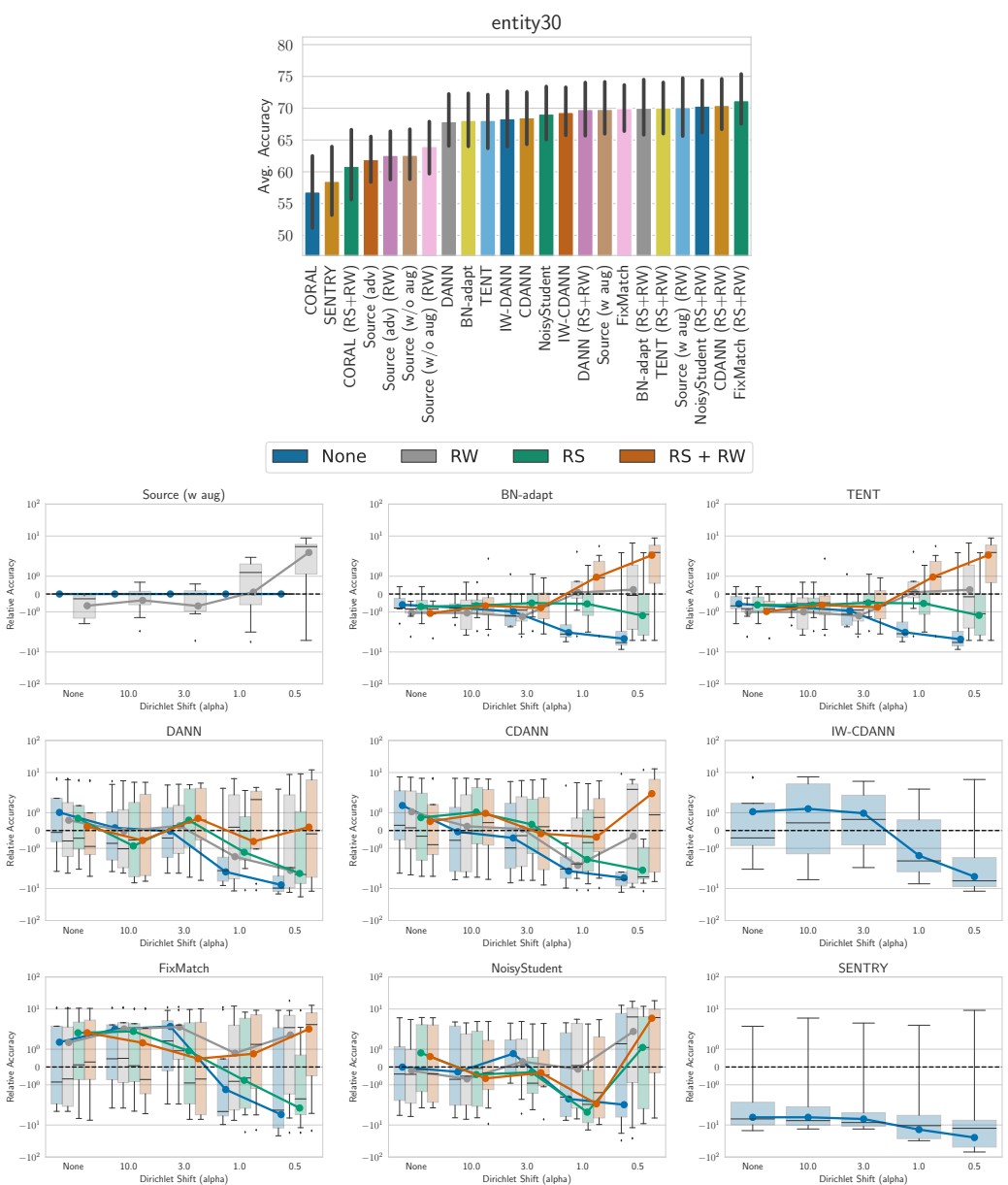

Figure 18: Entity30. Relative performance and accuracy plots for different DA algorithms across various shift pairs in Entity30.

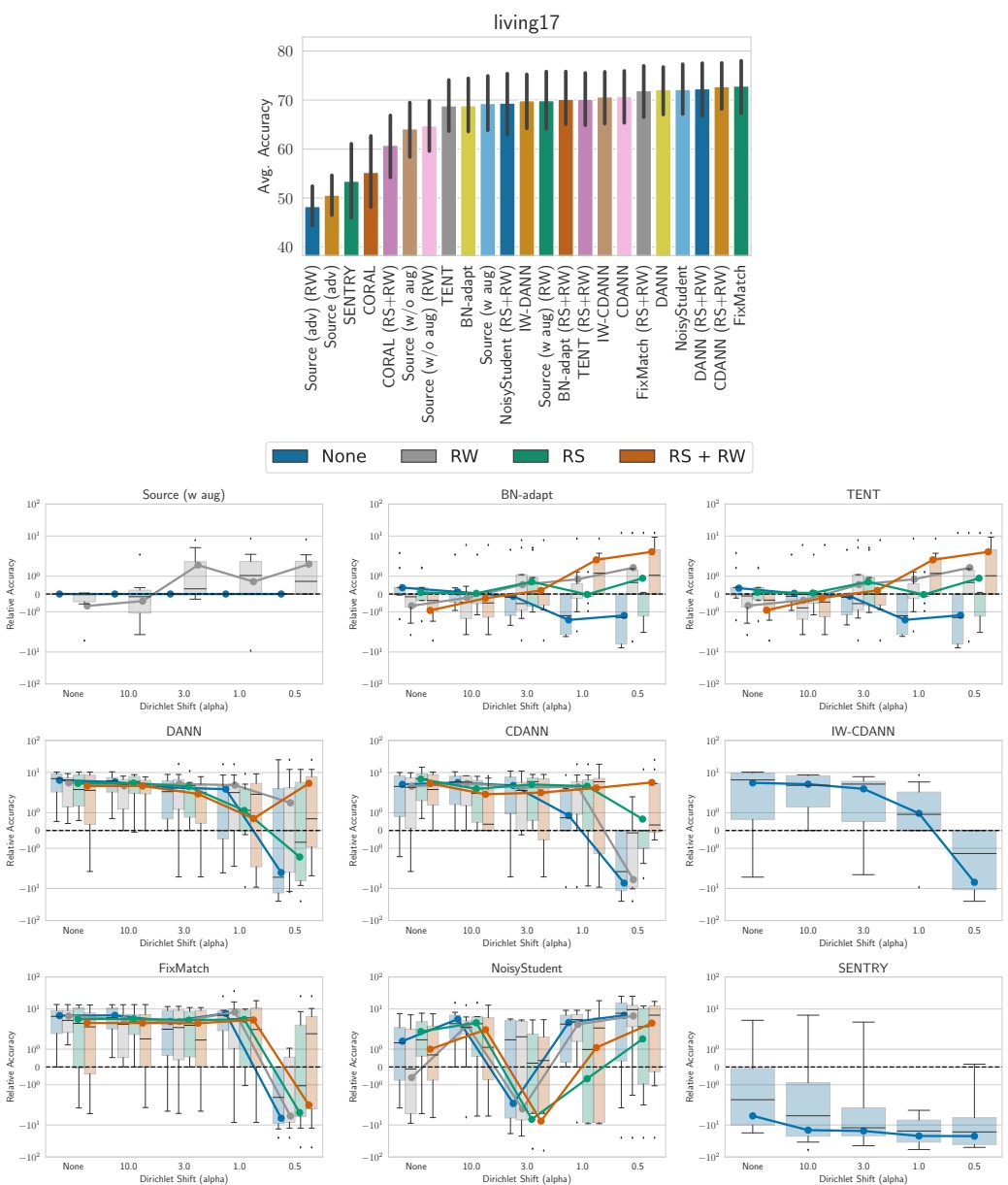

Figure 19: Living 17. Relative performance and accuracy plots for different DA algorithms across various shift pairs in Living17.

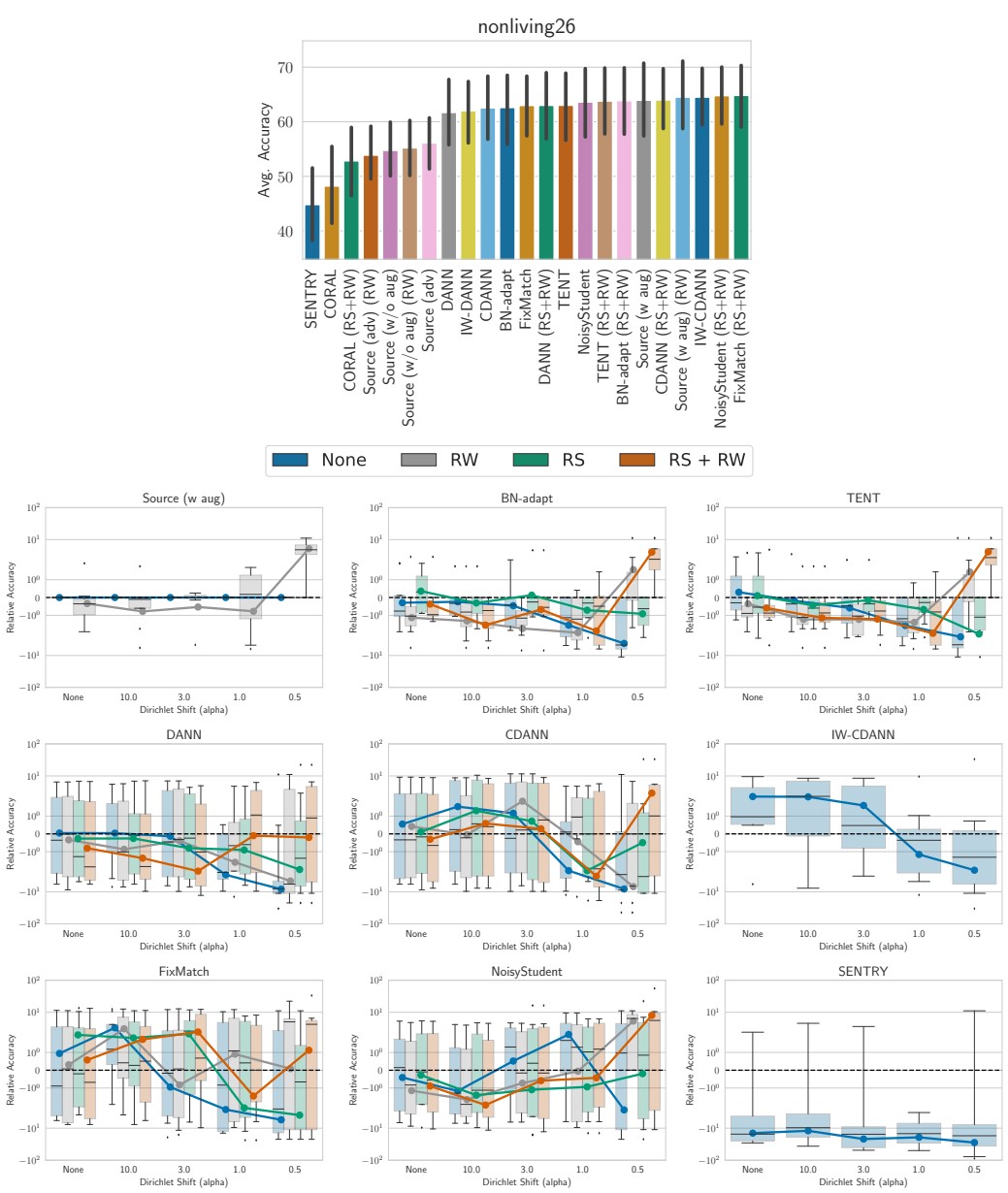

Figure 20: Nonliving 26. Relative performance and accuracy plots for different DA algorithms across various shift pairs in Nonliving26.

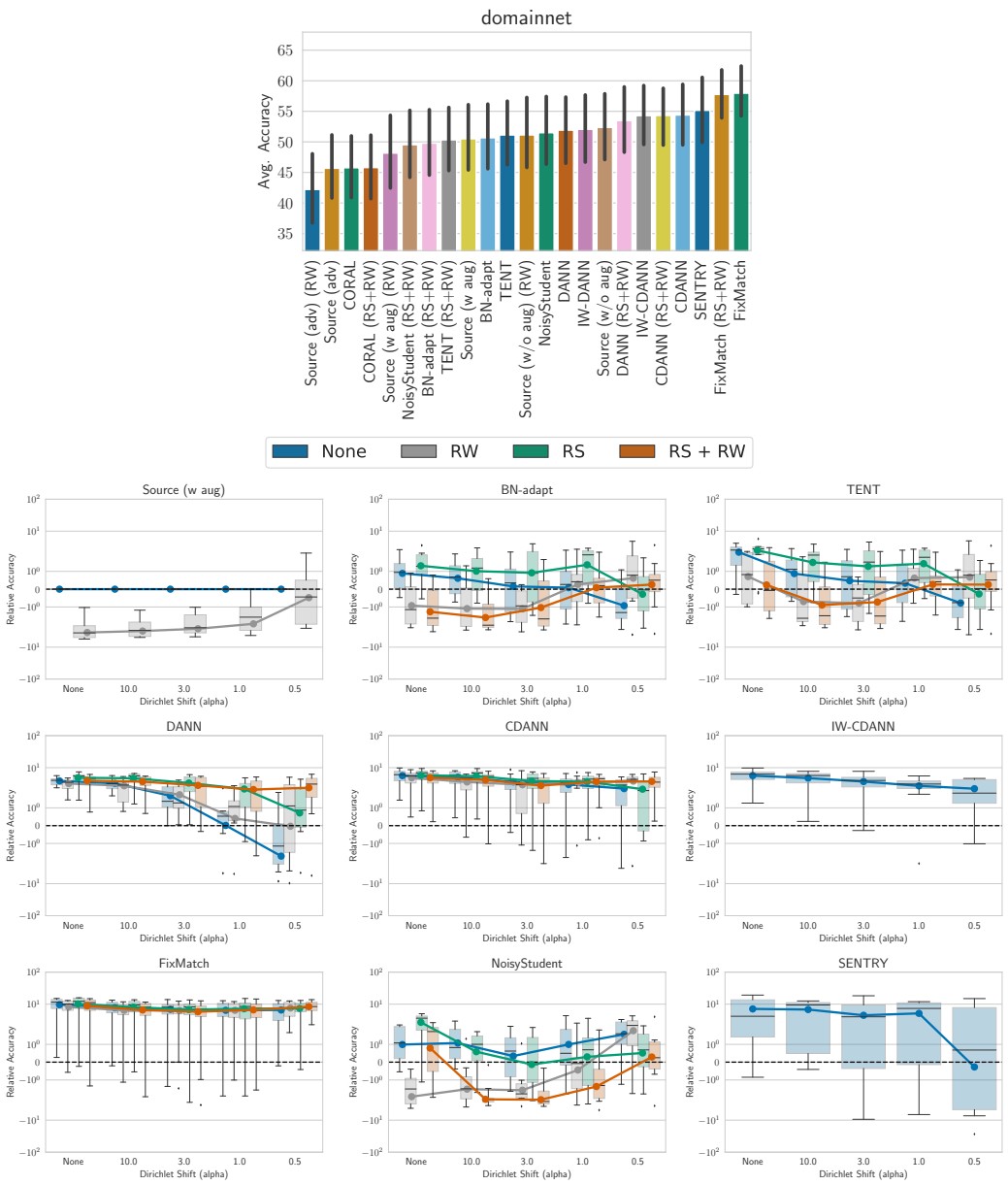

Figure 21: DomainNet. Relative performance and accuracy plots for different DA algorithms across various shift pairs in DomainNet.

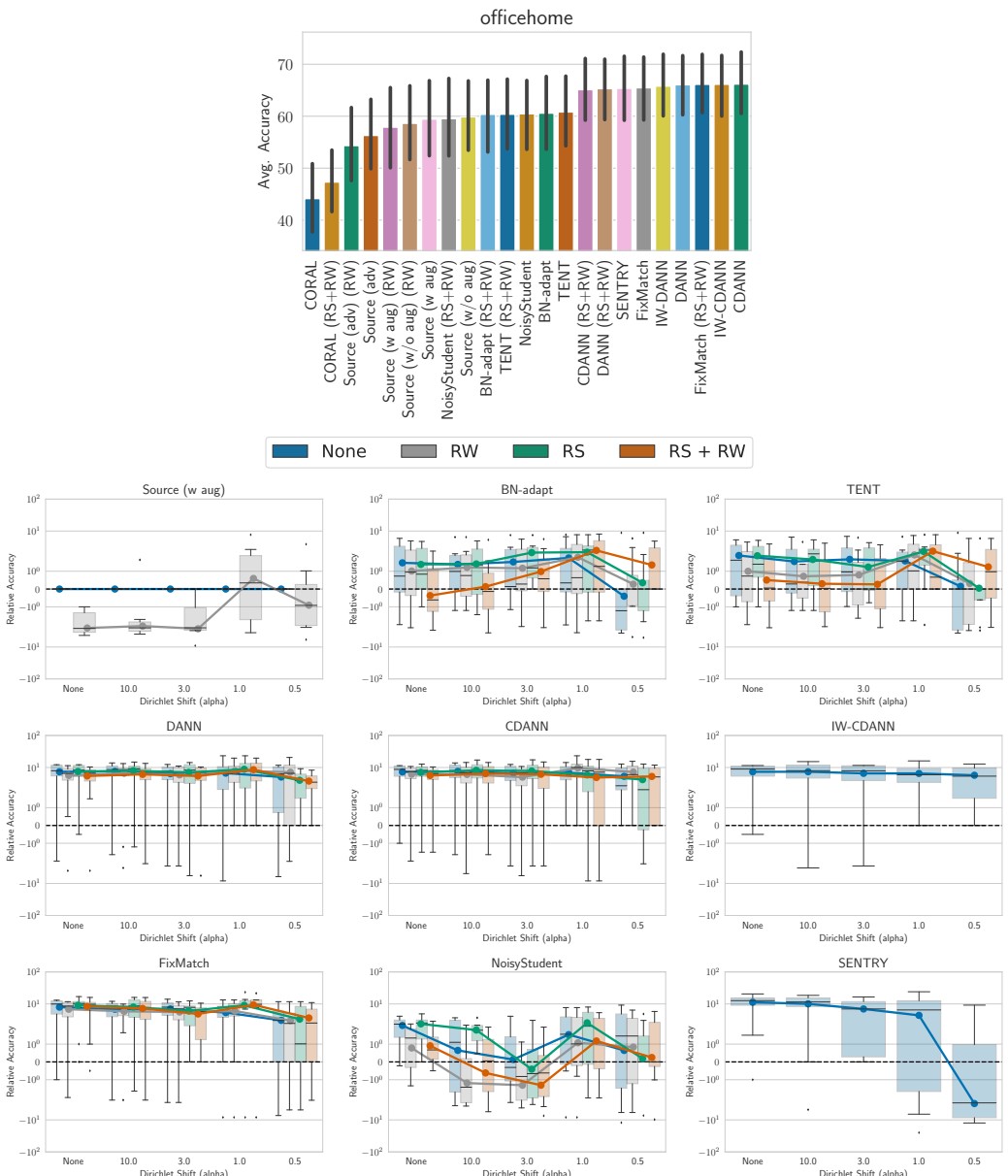

Figure 22: Officehome. Relative performance and accuracy plots for different DA algorithms across various shift pairs in Officehome.

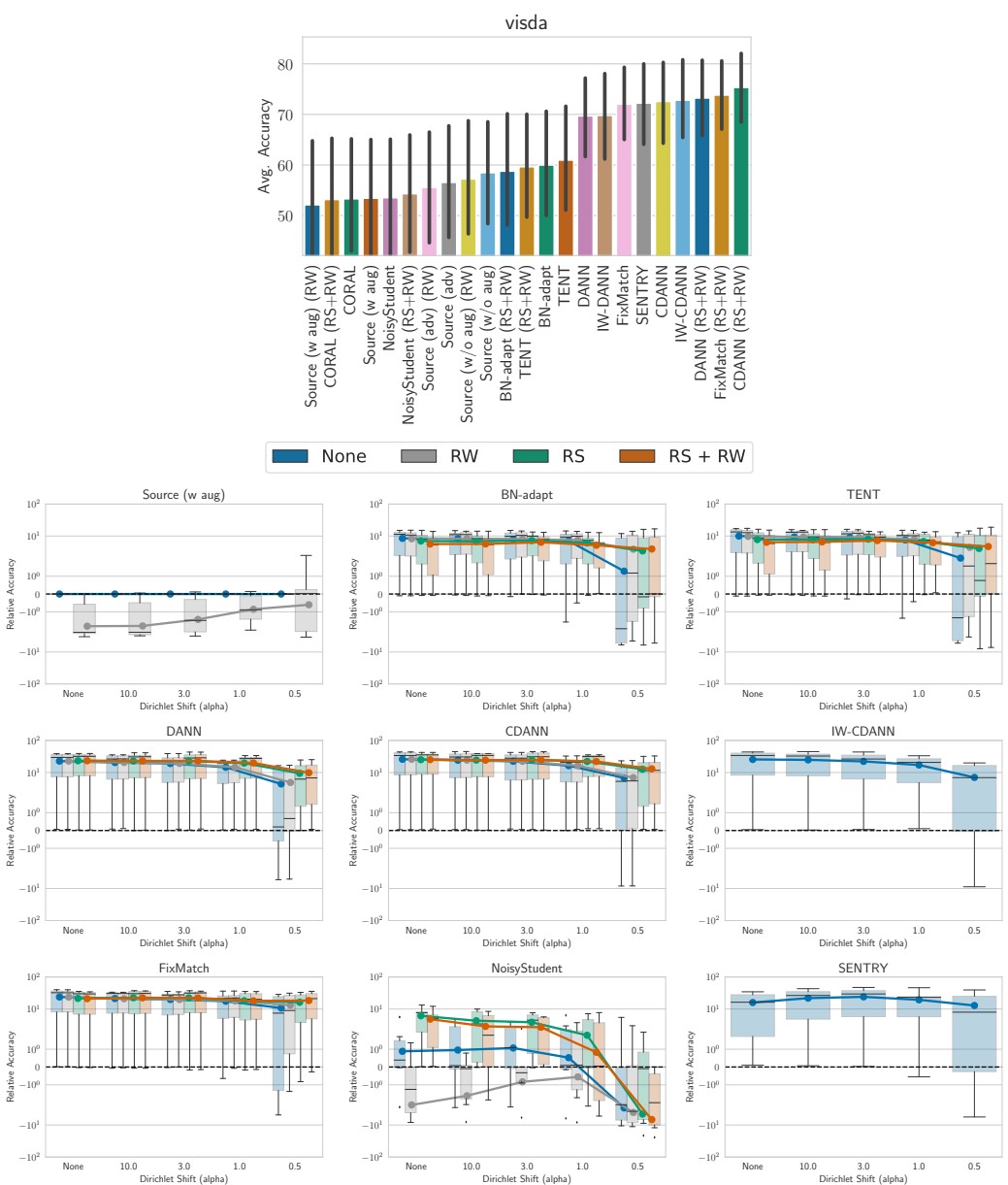

Figure 23: Visda. Relative performance and accuracy plots for different DA algorithms across various shift pairs in Visda.

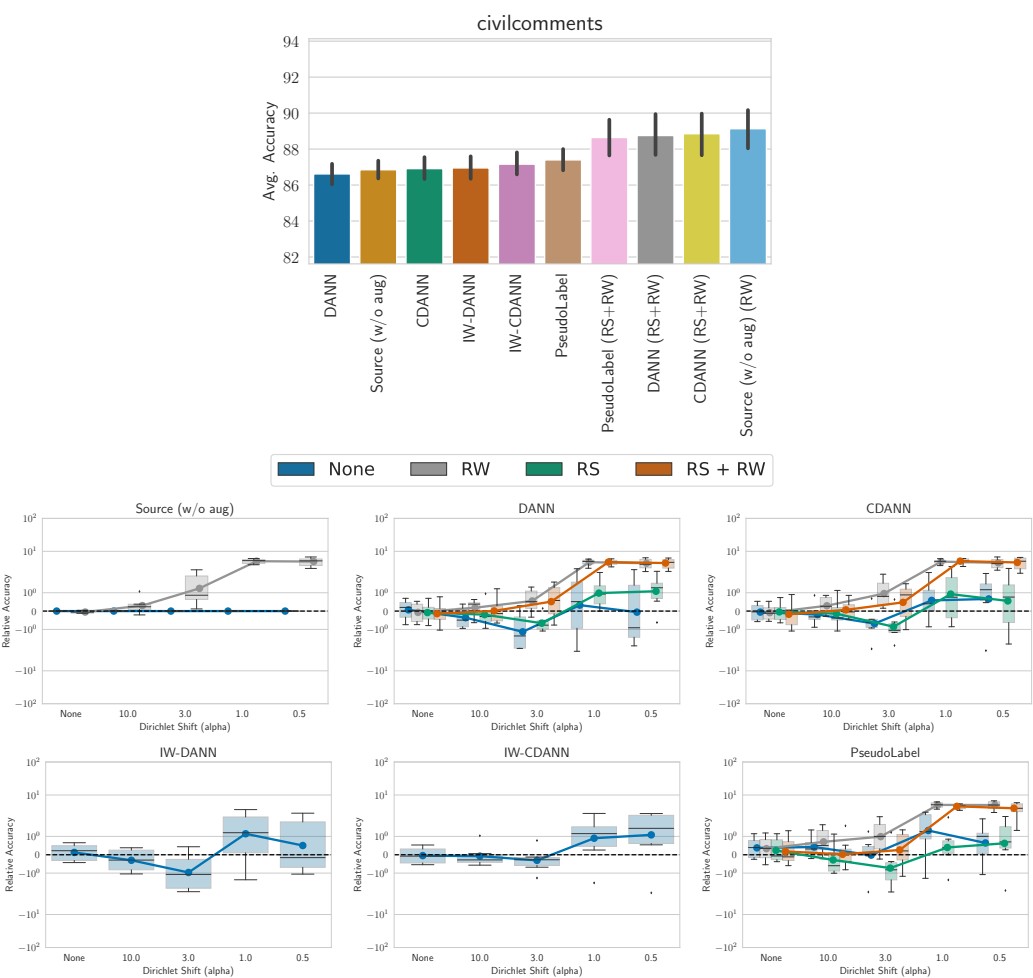

Figure 24: Civilcomments. Relative performance and accuracy plots for different DA algorithms across various shift pairs in Civilcomments.

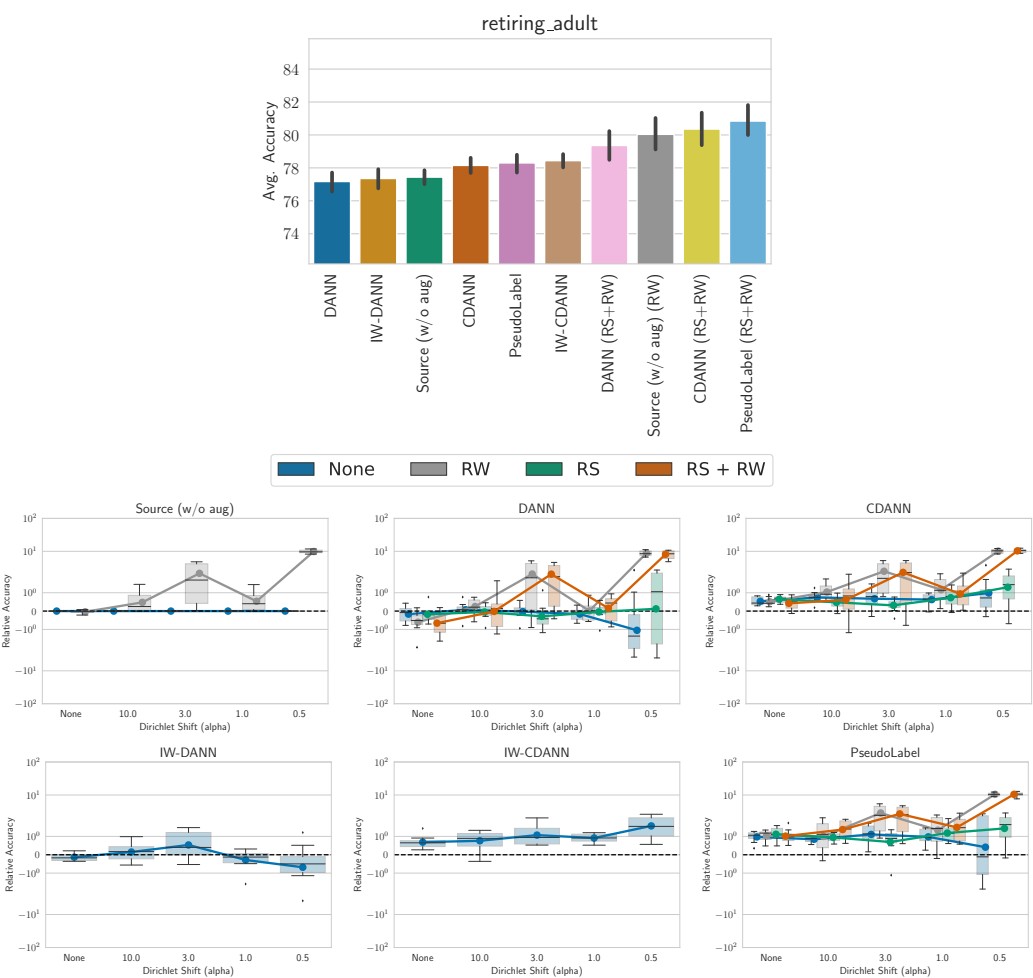

Figure 25: Retiring Adults. Relative performance and accuracy plots for different DA algorithms across various shift pairs in Retiring Adults.

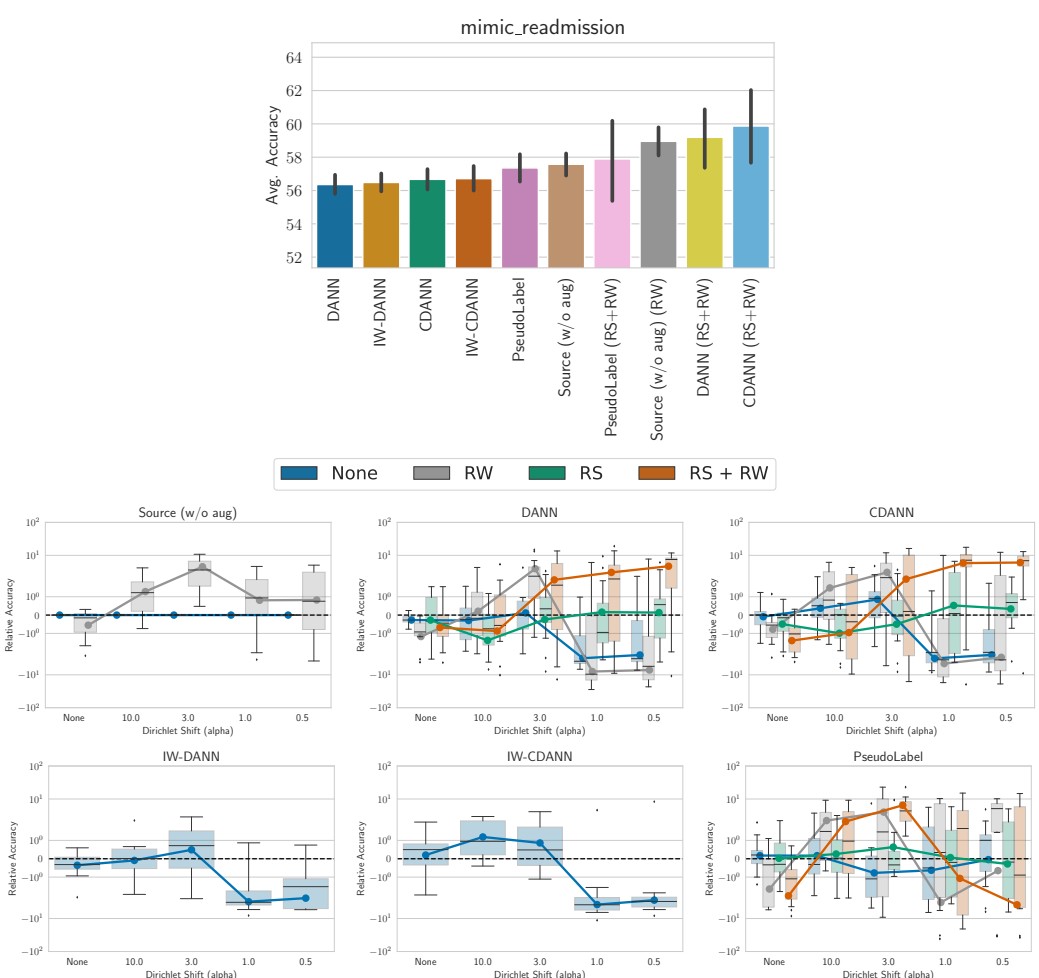

Figure 26: Mimic Readmissions. Relative performance and accuracy plots for different DA algorithms across various shift pairs in Mimic Readmissions.

# J AGGREGATE ACCURACY WITH DIFFERENT DA METHODS ON EACH DATASET

| Dataset | Source | DANN | IW-DANN | CDANN | IW-CDANN | PseudoLabel |
|---|---|---|---|---|---|---|
| Civilcomments | 86.85 | 86.62 | 86.95 | 86.91 | 87.16 | 87.4 |

| Dataset | Source | | DANN | | | | CDANN | | | | PseudoLabel | | | |
|---|---|---|---|---|---|---|---|---|---|---|---|---|---|---|
| | None | RW | None | RW | RS | RS+RW | None | RW | RS | RS+RW | None | RW | RS | RS+RW |
| Civilcomments | 86.8 | 89.1 | 86.6 | 88.8 | 87.1 | 88.8 | 86.9 | 89.0 | 86.9 | 88.9 | 87.4 | 89.3 | 86.9 | 88.6 |

Table 2: *Results with different DA methods on NLP datasets aggregated across target label marginal shifts.*

| Dataset | Source | DANN | IW-DANN | CDANN | IW-CDANN | PseudoLabel |
|---|---|---|---|---|---|---|
| Retiring Adult | 77.44 | 77.17 | 77.35 | 78.15 | 78.44 | 78.30 |
| Mimic Readmission | 57.57 | 56.36 | 56.48 | 56.67 | 56.71 | 57.35 |

| Dataset | Source | | DANN | | | | CDANN | | | | PseudoLabel | | | |
|---|---|---|---|---|---|---|---|---|---|---|---|---|---|---|
| | None | RW | None | RW | RS | RS+RW | None | RW | RS | RS+RW | None | RW | RS | RS+RW |
| Retiring Adults | 77.4 | 80.0 | 77.2 | 79.5 | 77.4 | 79.4 | 78.1 | 80.5 | 78.1 | 80.4 | 78.3 | 80.8 | 78.5 | 80.8 |
| Mimic Readmissions | 57.6 | 59.0 | 56.4 | 55.1 | 57.3 | 59.2 | 56.7 | 56.8 | 57.4 | 59.9 | 57.4 | 57.7 | 57.7 | 57.9 |

Table 3: *Results with different DA methods on tabular datasets aggregated across target label marginal shifts.*

| Dataset | Source (wo aug) | Source (w aug) | BN-adapt | TENT | DANN | IW-DAN | CDAN | IW-CDAN | Fix-Match | Noisy-Student | Sentry |
|---|---|---|---|---|---|---|---|---|---|---|---|
| CIFAR-10 | 89.69 | 89.14 | 89.21 | 89.20 | 90.86 | 90.78 | 90.00 | 89.93 | 91.87 | 90.72 | 91.83 |
| CIFAR-100 | 65.99 | 76.69 | 77.57 | 77.58 | 74.80 | 74.81 | 74.57 | 74.66 | 79.03 | 77.60 | 74.74 |
| FMoW | 64.00 | 68.99 | 65.52 | 66.55 | 60.11 | 60.33 | 60.79 | 61.05 | 68.37 | 68.90 | 51.06 |
| Camelyon | 77.42 | 76.95 | 85.70 | 82.48 | 86.66 | 85.89 | 85.45 | 84.27 | 86.29 | 79.29 | 86.81 |
| Domainnet | 52.37 | 50.50 | 50.66 | 51.12 | 51.91 | 52.05 | 54.40 | 54.29 | 57.96 | 51.49 | 55.16 |
| Entity13 | 76.93 | 80.07 | 77.99 | 78.04 | 78.26 | 78.75 | 79.74 | 79.28 | 80.25 | 80.37 | 73.58 |
| Entity30 | 62.61 | 69.83 | 68.09 | 68.09 | 67.90 | 68.36 | 68.51 | 69.34 | 69.95 | 69.10 | 58.51 |
| Living17 | 64.13 | 69.30 | 68.84 | 68.82 | 72.12 | 69.87 | 70.72 | 70.65 | 72.86 | 72.16 | 53.44 |
| Nonliving26 | 54.75 | 63.95 | 62.60 | 63.02 | 61.69 | 61.99 | 62.53 | 64.51 | 62.98 | 63.60 | 44.82 |
| Officehome | 59.89 | 59.45 | 60.59 | 60.82 | 66.05 | 65.79 | 66.19 | 66.15 | 65.48 | 60.47 | 65.37 |
| Visda | 58.47 | 53.41 | 59.98 | 60.96 | 69.69 | 69.79 | 72.55 | 72.80 | 72.02 | 53.51 | 72.23 |
| **Avg** | 66.02 | 68.94 | 69.70 | 69.70 | 70.92 | 70.77 | 71.40 | 71.54 | 73.37 | 69.75 | 66.14 |

Table 4: *Results with different DA methods on vision datasets aggregated across target label marginal shifts.* While no single DA method performs consistently across different datasets, FixMatch seems to provide the highest aggregate improvement over a source-only classifier in our testbed.

| Dataset | Source | | BN-adapt | | | | CDANN | | | | FixMatch | | | |
|---|---|---|---|---|---|---|---|---|---|---|---|---|---|---|
| | None | RW | None | RW | RS | RS+RW | None | RW | RS | RS+RW | None | RW | RS | RS+RW |
| CIFAR-10 | 89.1 | 89.4 | 89.2 | 91.4 | 92.1 | 92.9 | 90.0 | 91.3 | 91.4 | 92.5 | 91.9 | 93.1 | 93.6 | 94.1 |
| CIFAR-100 | 76.7 | 77.5 | 77.6 | 78.8 | 77.9 | 79.0 | 74.6 | 75.8 | 74.1 | 75.3 | 79.0 | 79.6 | 79.1 | 79.8 |
| FMoW | 69.0 | 70.3 | 65.5 | 67.2 | 66.2 | 65.6 | 60.8 | 61.9 | 57.0 | 55.2 | 68.4 | 69.4 | 64.9 | 66.7 |
| Camelyon | 77.0 | 77.9 | 85.7 | 85.9 | 88.5 | 89.3 | 85.5 | 85.8 | 87.9 | 88.5 | 86.3 | 87.0 | 86.6 | 86.8 |
| Domainnet | 50.5 | 48.2 | 50.7 | 50.1 | 51.4 | 49.8 | 54.4 | 54.2 | 54.7 | 54.3 | 58.0 | 57.5 | 58.4 | 57.8 |
| Entity13 | 80.1 | 80.9 | 78.0 | 79.4 | 79.8 | 80.7 | 79.7 | 80.2 | 80.6 | 81.4 | 80.3 | 81.9 | 81.4 | 82.4 |
| Entity30 | 69.8 | 70.1 | 68.1 | 69.2 | 69.1 | 70.0 | 68.5 | 69.6 | 69.4 | 70.5 | 70.0 | 71.6 | 70.1 | 71.2 |
| Living17 | 69.3 | 69.9 | 68.8 | 69.7 | 69.6 | 70.1 | 70.7 | 71.3 | 72.9 | 72.7 | 72.9 | 72.8 | 72.3 | 71.9 |
| Nonliving26 | 63.9 | 64.5 | 62.6 | 63.0 | 63.7 | 63.9 | 62.5 | 62.9 | 63.8 | 64.0 | 63.0 | 64.7 | 63.9 | 64.8 |
| Officehome | 59.4 | 57.9 | 60.6 | 60.5 | 60.9 | 60.4 | 66.2 | 66.3 | 66.1 | 65.1 | 65.5 | 64.9 | 66.5 | 66.1 |
| Visda | 53.4 | 52.1 | 60.0 | 60.6 | 59.5 | 58.8 | 72.6 | 72.6 | 75.3 | 75.3 | 72.0 | 72.5 | 73.5 | 73.8 |
| **Avg** | 68.9 | 69.0 | 69.7 | 70.5 | 70.8 | 70.9 | 71.4 | 72.0 | 72.1 | 72.3 | 73.4 | 74.1 | 73.7 | 74.1 |

| Dataset | TENT | | | | DANN | | | | NoisyStudent | | | |
|---|---|---|---|---|---|---|---|---|---|---|---|---|
| | None | RW | RS | RS+RW | None | RW | RS | RS+RW | None | RW | RS | RS+RW |
| CIFAR-10 | 89.2 | 91.4 | 92.1 | 92.9 | 90.9 | 92.3 | 91.5 | 92.6 | 90.7 | 90.8 | 90.6 | 90.7 |
| CIFAR-100 | 77.6 | 78.8 | 78.0 | 79.0 | 74.8 | 75.9 | 74.8 | 76.1 | 77.6 | 78.0 | 77.9 | 78.0 |
| FMoW | 66.6 | 67.4 | 66.7 | 66.1 | 60.1 | 61.6 | 56.4 | 54.5 | 68.9 | 69.8 | 67.1 | 68.0 |
| Camelyon | 82.5 | 82.7 | 87.8 | 88.9 | 86.7 | 87.3 | 88.4 | 88.8 | 79.3 | 79.1 | 79.2 | 79.3 |
| Domainnet | 51.1 | 50.6 | 51.8 | 50.3 | 51.9 | 52.1 | 53.6 | 53.5 | 51.5 | 49.8 | 51.3 | 49.5 |
| Entity13 | 78.0 | 79.5 | 79.8 | 80.8 | 78.3 | 79.4 | 79.7 | 80.8 | 80.4 | 81.5 | 80.6 | 81.7 |
| Entity30 | 68.1 | 69.2 | 69.1 | 70.1 | 67.9 | 69.2 | 69.0 | 69.8 | 69.1 | 70.1 | 69.3 | 70.3 |
| Living17 | 68.8 | 69.7 | 69.6 | 70.1 | 72.1 | 73.0 | 71.8 | 72.3 | 72.2 | 71.1 | 69.3 | 69.4 |
| Nonliving26 | 63.0 | 63.4 | 63.3 | 63.8 | 61.7 | 62.4 | 63.1 | 63.0 | 63.6 | 64.3 | 63.2 | 64.8 |
| Officehome | 60.8 | 60.4 | 60.9 | 60.4 | 66.1 | 66.1 | 66.5 | 65.3 | 60.5 | 59.5 | 60.8 | 59.5 |
| Visda | 61.0 | 61.5 | 60.3 | 59.6 | 69.7 | 69.9 | 73.1 | 73.2 | 53.5 | 51.5 | 55.7 | 54.3 |
| **Avg** | 69.7 | 70.4 | 70.8 | 71.1 | 70.9 | 71.7 | 71.6 | 71.8 | 69.7 | 69.6 | 69.5 | 69.6 |

Table 5: *Results with DA methods paired with re-sampling (RS) and re-weighting (RW) correction (with RLLS estimate) aggregated across target label marginal shifts for vision datasets.* RS and RW seem to help for all datasets and they both together significantly improve aggregate performance over no correction for all DA methods.

## K    DESCRIPTION OF DEEP DOMAIN ADAPTATION METHODS

In this section, we summarize deep DA methods compared in our RLSBENCH testbed. We also discuss how each method combines with our meta-algorithm to handle shift in class proportion.

### K.1    SOURCE ONLY TRAINING

As a baseline, we consider empirical risk minimization on the labeled source data. Since this simply ignores the unlabeled target data, we call this as source only training. As mentioned in the main paper, we perform source only training with and without data augmentations. Formally, we minimize the following ERM loss:

$$L_{\text{source only}}(f) = \frac{1}{n} \sum_{i=1}^{n} \ell(f(T(x_i), y_i)), \tag{4}$$

where $T$ is the stochastic data augmentation operation for vision datasets and $\ell$ is a loss function. For NLP and tabular datasets, $T$ is the identity function. Throughout the paper, we use cross-entropy loss minimization. Unless specified otherwise, we use strong augmentations as the data augmentation technique for vision datasets. For NLP and tabular datasets, we do not use any data augmentation.

As mentioned in the main paper, we do not include re-sampling results with a source only model as it is trained only on source data and we observed no differences with just balancing the source data (as for most datasets source is already balanced) in our experiments. After obtaining a classifier $f$, we can first estimate the target label marginal and then adjust the classifier $f$ with post-hoc re-weighting with importance ratios $w_t(y) = \hat{p}_t(y)/\hat{p}_s(y)$.

**Adversarial training of a source only model**    Along with standard training of a source only model with data augmentation, we experiment with adversarially robust models (Madry et al., 2017). To train adversarially robust models, we replace the standard ERM objective with a robust risk minimization objective:

$$L_{\text{source only (adv)}}(f) = \frac{1}{n} \sum_{i=1}^{n} \ell(R(T(x_i), y_i), y_i), \tag{5}$$

where $R(\cdot)$ performs the adversarial augmentation. In our paper, we use targeted Projected Gradient Descent (PGD) attacks with $\ell_2$ perturbation model.

### K.2    DOMAIN-ADVERSARIAL TRAINING METHODS

Domain-adversarial trianing methods seek to learn feature representations that are invariant across domains. These methods aimed at practical problems with non-overlapping support and are motivated by theoretical results showing that the gap between in- and out-of-distribution performance depends on some measure of divergence between the source and target distributions (Ben-David et al., 2010a; Ganin et al., 2016). While simultaneously minimizing the source error, these methods align the representations between source and target distribution. To perform alignment, these methods penalize divergence between feature representations across domains, encouraging the model to produce feature representations that are similar across domain.

Before describing these methods, we first define some notation. Consider a model $f = g \circ h$, where $h : \mathcal{X} \to \mathbb{R}^d$ is the featurizer that maps the inputs to some $d$ dimensional feature space, and the head $g : \mathbb{R}^d \to \Delta^{k-1}$ maps the features to the prediction space. Following Sagawa et al. (2021), with all of our domain invariant methods, we use strong augmentations with source and target data for vision datasets. For NLP and tabular datasets, we do not use any data augmentation.

**DANN**    DANN was proposed in Ganin et al. (2016). DANN approximates the divergence between feature representations of source and target domain by leveraging a domain discriminator classifier. Domain discriminator $f_d$ aims to discriminate between source and target domains. Given a batch of inputs from source and target, this deep network $f_d$ classifies whether the examples are from the source data or target data. In particular, the following loss function is used:

$$L_{\text{domain disc.}}(f_d) = \frac{1}{n} \sum_{i=1}^{n} \ell(f_d(h(T(x_i))), 0) + \frac{1}{m} \sum_{i=n+1}^{n+m} \ell(f_d(h(T(x_i))), 1), \tag{6}$$

where $\{x_1, x_2, \ldots, x_n\}$ are $n$ source examples and $\{x_{n+1}, \ldots, x_{m+n}\}$ are $m$ target examples. Overall, the following loss function is used to optimize models with DANN:

$$L_{\text{DANN}}(h, g, f_d) = L_{\text{source only}}(g \circ h) - \lambda L_{\text{domain disc.}}(f_d) \,. \tag{7}$$

$L_{\text{DANN}}(h, g, f_d)$ is maximized with respect to the domain discriminator classifier and $L_{\text{DANN}}(h, g, f_d)$ minimized with respect to the underlying featurize and the source classifier. This is achieved by gradient reversal layer in practice. To train, three networks, we use three different learning rate $\eta_f, \eta_g$, and $\eta_{f_d}$. We discuss these hyperparameter details in App. L. We adapted our DANN implementation from Sagawa et al. (2021) and Transfer learning library (Jiang et al., 2022).

**CDANN**   Conditional Domain adversarial neural network is a variant of DANN (Long et al., 2018). Here the domain discriminator is conditioned on the classifier $g$'s prediction. In particular, instead of training the domain discriminator on the representation output of $h$, these methods operate on the outer product between the feature presentation $h(x)$ at an input $x$ and the classifier's probabilistic prediction $f = g \circ h(x)$ (i.e., $h(x) \otimes f(x)$). Thus instead of training the domain discriminator classifier $f_d$ on the $d$ dimensional input space, they train it on $d \times k$ dimensional space. In particular, the following loss function is used:

$$L_{\text{CDAN domain disc.}}(f_d, g, h) = \frac{1}{n} \sum_{i=1}^{n} \ell(f_d(f \otimes h(T(x_i))), 0) + \frac{1}{n} \sum_{i=n+1}^{n+m} \ell(f_d(f \otimes h(T(x_i))), 1) \,, \tag{8}$$

where $\{x_1, x_2, \ldots, x_n\}$ are $n$ source examples and $\{x_{n+1}, \ldots, x_{m+n}\}$ are $m$ target examples. The overall loss is the same as DANN where $L_{\text{domain disc.}}(f_d)$ is replaced with $L_{\text{CDAN domain disc.}}(f_d, g, h)$.

We adapted our implementation for CDANN from Transfer learning library (Jiang et al., 2022).

To adapt DANN and CDANN to our meta algorithm, at each epoch we can perform re-balancing of source and target data as in Step 1 and 4 of Algorithm 1. After obtaining the classifier $f$, we can use this classifier to first obtain an estimate of the target label marginal and then perform re-weighting adjustment with the obtained estimate.

**IW-DANN and IW-CDANN**   Tachet et al. (2020) proposed training with importance re-weighting correction with DANN and CDANN objectives to accommodate for the shift in the target label proportion. In particular, at every epoch of training they first estimate the importance ratio $\widehat{w}_t$ (with BBSE on training source and training target data) and then re-weight the domain discriminator objective and ERM objective. In particular, the domain discriminator loss for IW-DANN can be written as:

$$L_{\text{domain disc.}}^{\widehat{w}}(f_d) = \frac{1}{n} \sum_{i=1}^{n} \widehat{w}(y_i) \ell(f_d(h(T(x_i))), 0) + \frac{1}{n} \sum_{i=n+1}^{n+m} \ell(f_d(h(T(x_i))), 1) \,, \tag{9}$$

where we multiply the source loss with importance weights. Similarly, we can re-write the source only training objective with importance re-weighting as follows:

$$L_{\text{source only}}^{\widehat{w}}(f) = \frac{1}{n} \sum_{i=1}^{n} \widehat{w}(y_i) \ell(f(T(x_i), y_i)) \,. \tag{10}$$

Overall, the following objective is used to optimize models with IW-DANN:

$$L_{\text{IW-DANN}}(h, g, f_d) = L_{\text{source only}}^{\widehat{w}}(g \circ h) - \lambda L_{\text{domain disc.}}^{\widehat{w}}(f_d) \,, \tag{11}$$

where the importance weights are updated after every epoch with classifier obtained in previous step. Similarly, with using importance re-weights with the CDANN objective, we obtain IW-CDANN objective.

In population, IW-CDANN and IW-DANN correction matches the correction with our meta-algorithm for DANN and CDANN. However, the behavior this importance re-weighting correction can be different from our meta-algorithm for over-parameterized models with finite data (Byrd & Lipton, 2019). Recent empirical and theoretical findings have highlighted that importance re-weighting have

minor to no effect on overparameterized models when trained for several epochs (Byrd & Lipton, 2019; Xu et al., 2021). On the other hand, with finite samples, re-sampling (when class labels are available) has shown different and promising empirical behavior (An et al., 2020; Idrissi et al., 2022). This may highlight the differences in the behavior of IW-CDANN (or IW-DANN) with our meta algorithm on CDANN (or DANN).

We refer to the implementation provided by the authors (Tachet et al., 2020).

### K.3 SELF-TRAINING METHODS

Self-training methods leverage unlabeled data by 'pseudo-labeling' unlabeled examples with the classifier's own predictions and training on them as if they were labeled examples. Recent self-training methods also often make use of consistency regularization, for example, encouraging the model to make similar predictions on augmented versions of unlabeled example. In our work, we experiment with the following methods:

**PseudoLabel** (Lee et al., 2013) proposed PseudoLabel that leverages unlabeled examples with classifier's own prediction. This algorithm dynamically generates psuedolabels and overfits on them in each batch. In particular, while pseudolabels are generated on unlabeled examples, the loss is computed with respect to the same label. PseudoLabel only overfits to the assigned label if the confidence of the prediction is greater than some threshold $\tau$.

Refer to $T$ as the data-augmentation technique (i.e., identity for NLP and tabular datasets and strong augmentation for vision datasets). Then, PseudoLabel uses the following loss function:

$$L_{\text{PseudoLabel}}(f) = \frac{1}{n} \sum_{i=1}^{n} \ell(f(T(x_i), y_i)) + \frac{\lambda_t}{m} \sum_{i=n+1}^{m+n} \ell(f(T(x_i), \widetilde{y}_i)) \cdot \mathbb{I}\left[\max_y f_y(T(x_i)) \geqslant \tau\right],$$

where $\widetilde{y}_i = \arg\max_y f_y(T(x_i))$. PseudoLabel increases $\lambda_t$ between labeled and unlabeled losses over epochs, initially placing 0 weight on unlabeled loss and then linearly increasing the unlabeled loss weight until it reaches the full value of hyperparameter $\lambda$ at some threshold step. We fix the step at which $\lambda_t$ reaches its maximum value $\lambda$ be 40% of the total number of training steps, matching the implementation to (Sohn et al., 2020; Sagawa et al., 2021).

**FixMatch** Sohn et al. (2020) proposed FixMatch as a variant of the simpler Pseudo-label method (Lee et al., 2013). This algorithm dynamically generates psuedolabels and overfits on them in each batch. FixMatch employs consistency regularization on the unlabeled data. In particular, while pseudolabels are generated on a weakly augmented view of the unlabeled examples, the loss is computed with respect to predictions on a strongly augmented view. The intuition behind such an update is to encourage a model to make predictions on weakly augmented data consistent with the strongly augmented example. Moreover, FixMatch only overfits to the assigned labeled with weak augmentation if the confidence of the prediction with strong augmentation is greater than some threshold $\tau$.

Refer to $T_{\text{weak}}$ as the weak-augmentation and $T_{\text{strong}}$ as the strong-augmentation function. Then, FixMatch uses the following loss function:

$$\begin{aligned} L_{\text{FixMatch}}(f) = &\frac{1}{n} \sum_{i=1}^{n} \ell(f(T_{\text{strong}}(x_i), y_i)) \\ &+ \frac{\lambda}{m} \sum_{i=n+1}^{m+n} \ell(f(T_{\text{strong}}(x_i), \widetilde{y}_i)) \cdot \mathbb{I}\left[\max_y f_y(T_{\text{strong}}(x_i)) \geqslant \tau\right], \end{aligned}$$

where $\widetilde{y}_i = \arg\max_y f_y(T_{\text{weak}}(x_i))$. We adapted our implementation from Sagawa et al. (2021) which matches the implementation of Sohn et al. (2020) except for one detail. While Sohn et al. (2020) augments labeled examples with weak augmentation, Sagawa et al. (2021) proposed to strongly augment the labeled source examples.

**NoisyStudent** Xie et al. (2020) proposed a different variant of Pseudo-labeling. Unlike FixMatch, Noisy Student generates pseudolabels, fixes them, and then trains the model until convergence before generating new pseudolabels. The first set of pseudolabels are obtained by training an initial teacher

model only on the source labeled data. Then in each iteration, randomly initialized models fit the labeled source data and pseudolabeled target data with pseudolabels assigned by the converged model in the previous iteration. Noisy student objective can be summarized as:

$$L_{\text{NoisyStudent}}(f^N) = \frac{1}{n} \sum_{i=1}^{n} \ell(f^N(T_{\text{strong}}(x_i), y_i)) + \frac{1}{m} \sum_{i=n+1}^{m+n} \ell(f^N(T_{\text{strong}}(x_i), \widetilde{y}_i)),$$

where $\widetilde{y}_i = \arg\max_y f_y^{N-1}(T_{\text{weak}}(x_i))$ is computed with the classifier obtained at $N-1$ step. Note that the randomly initialized model at each iteration uses a dropout of $p = 0.5$ in the penultimate layer. We adopted our implementation of NoisyStudent to Sagawa et al. (2021). To initialize the initial teacher model, we use the source-only model trained with strong augmentations without dropout.

**SENTRY**    Prabhu et al. (2021) proposed a different variant of pseudolabeling method. This method is aimed to tackle DA under relaxed label shift scenario. a SENTRY incorporates a target instance based on its predictive consistency under a committee of strong image transformations. In particular, SENTRY makes N strong augmentations of an unlabeled target example and makes a prediction on those. If the majority of the committee matches the prediction on the sample example with weak-augmentation then entropy is minimized on that example, otherwise the entropy is maximized. Moreover, the authors employ an 'information-entropy' objective aimed to match the prediction at every example with the estimated target label marginal. Overall the SENTRY objective is defined as follows:

$$L_{\text{SENTRY}}(f) = \frac{1}{n} \sum_{i=1}^{n} \ell(f(T_{\text{strong}}(x_i), y_i)) + \frac{1}{m} \sum_{i=n+1}^{m+n} \sum_{j=1}^{k} f_k(y = j|x_i) \log(\widetilde{p}_t(y = j))$$

$$+ \lambda_{\text{unsup}} \frac{1}{m} \sum_{i=n+1}^{m+n} \sum_{j=1}^{k} -f_k(y = j|x_i) \log(f_k(y = j|x_i)) \cdot (2l(x) - 1),$$

where $l(x) \in \{0, 1\}$ is majority vote output of the committee consistency. For more details, we refer the reader to Prabhu et al. (2021). Additionally, at each training epoch, SENTRY balances the source data and pseudo-balances the target data. We adopted our implementation with the official implementation in Prabhu et al. (2021) with minor differences.

Since Fix-Match, NoisyStuent, and Sentry use strong data-augmentations in their implementation, the applicability of these algorithms is restricted to vision datasets. For NLP and tabular datasets, we only train models with PseudoLabel as it doesn't rely on any augmentation technique.

### K.4  TEST-TIME TRAINING METHODS

These take an already trained source model and adapt a few parameters (e.g. batch norm parameters, batch norm statistics) on the unlabeled target data with an aim to improve target performance. Hence, we restrict these methods to vision datasets with architectures that use batch norm. These methods are computationally cheaper than other DA methods in the suite as they adapt a classifier on-the-fly. We include the following methods in our experimental suite:

**BN-adapt**    Li et al. (2016) proposed batch norm adaptation. More recently, Schneider et al. (2020) showed gains with BN-adapt on common corruptions benchmark. Batch norm adaptation is applicable for deep models with batch norm parameters. With this method we simply adapt the Batchnorm statistics, in particular, mean and std of each batch norm layer.

**TENT**    Wang et al. (2021) proposed optimizing batch norm parameters to minimize the entropy of the predictor on the unlabeled target data. In our implementation of TENT, we perform BN-adapt before learning batch norm parameters.

**CORAL**    Sun et al. (2016) proposed CORAL to adapt a model trained on the source to target by whitening the feature representations. In particular, say $\widehat{\Sigma}_s$ is the empirical covariance of the target data representations and $\Sigma_s$ is the empirical covariance of the source data representations, CORAL adjusts a linear layer $g$ on target by re-training the final layer on the outputs: $\Sigma_t^{1/2} \Sigma_s^{-1/2} h(x)$.

DARE (Rosenfeld et al., 2022) simplified the procedure and showed that this is equivalent to training a linear head $h$ on $\Sigma_s^{-1/2} h(x)$ and whitening target data representations with $\Sigma_t^{-1/2} h(x)$ before input to the classifier. We choose to implement the latter procedure as it is cheap to train a single classifier in multi-domain datasets.

With our meta-algorithm, before adapting the source-only classifier with test time adaptation methods, we use it to perform the re-sampling correction. After obtaining the adapted classifier, we estimate target label marginal and use it to adjust the classifier with re-weighting.

## L  HYPERPARAMETER AND ARCHITECTURE DETAILS

### L.1  ARCHITECTURE AND PRETRAINING DETAILS

For all datasets, we used the same architecture across different algorithms:

- CIFAR-10: Resnet-18 (He et al., 2016) pretrained on Imagenet
- CIFAR-100: Resnet-18 (He et al., 2016) pretrained on Imagenet
- Camelyon: Densenet-121 (Huang et al., 2017) *not* pretrained on Imagenet as per the suggestion made in (Koh et al., 2021)
- FMoW: Densenet-121 (Huang et al., 2017) pretrained on Imagenet
- BREEDs (Entity13, Entity30, Living17, Nonliving26): Resnet-18 (He et al., 2016) *not* pretrained on Imagenet as per the suggestion in (Santurkar et al., 2021). The main rationale is to avoid pre-training on the superset dataset where we are simulating sub-population shift.
- Officehome: Resnet-50 (He et al., 2016) pretrained on Imagenet
- Domainnet: Resnet-50 (He et al., 2016) pretrained on Imagenet
- Visda: Resnet-50 (He et al., 2016) pretrained on Imagenet
- Civilcomments: Pre-trained DistilBERT-base-uncased (Sanh et al., 2019)
- Retiring Adults: We use an MLP with 2 hidden layers and 100 hidden units in both of the hidden layer
- Mimic Readmissions: We use the transformer architecture described in Yao et al. (2022)[2]

Except for Resnets on CIFAR datasets, we used the standard pytorch implementation (Gardner et al., 2018). For Resnet on cifar, we refer to the implementation here: `https://github.com/kuangliu/pytorch-cifar`. For all the architectures, whenever applicable, we add antialiasing (Zhang, 2019). We use the official library released with the paper.

For imagenet-pretrained models with standard architectures, we use the publicly available models here: `https://pytorch.org/vision/stable/models.html`. For imagenet-pretrained models on the reduced input size images (e.g. CIFAR-10), we train a model on Imagenet on reduced input size from scratch. We include the model with our publicly available repository. For bert-based models, we use the publicly available models here: `https://huggingface.co/docs/transformers/`.

### L.2  HYPERPARAMETERS

First, we tune learning rate and $\ell_2$ regularization parameter by fixing batch size for each dataset that correspond to maximum we can fit to 15GB GPU memory. We set the number of epochs for training as per the suggestions of the authors of respective benchmarks. Note that we define the number of epochs as a full pass over the labeled training source data. We summarize learning rate, batch size, number of epochs, and $\ell_2$ regularization parameter used in our study in Table 6.

For each algorithm, we use the hyperparameters reported in the initial papers. For domain-adversarial methods (DANN and CDANN), we refer to the suggestions made in Transfer Learning Library (Jiang et al., 2022). We tabulate hyperparameters for each algorithm next:

---

[2]https://github.com/huaxiuyao/Wild-Time/.

| Dataset | Epoch | Batch size | $\ell_2$ regularization | Learning rate |
|---|---|---|---|---|
| CIFAR10 | 50 | 200 | 0.0001 (chosen from {0.0001, 0.001,1e-5, 0.0}) | 0.01 (chosen from {0.001, 0.01, 0.0001}) |
| CIFAR100 | 50 | 200 | 0.0001 (chosen from {0.0001, 0.001,1e-5, 0.0}) | 0.01 (chosen from {0.001, 0.01, 0.0001}) |
| Camelyon | 10 | 96 | 0.01 (chosen from {0.01, 0.001, 0.0001, 0.0}) | 0.03 (chosen from {0.003, 0.3, 0.0003, 0.03}) |
| FMoW | 30 | 64 | 0.0 (chosen from {0.0001, 0.001,1e-5,0.0}) | 0.0001 (chosen from {0.001, 0.01, 0.0001}) |
| Entity13 | 40 | 256 | 5e-5 (chosen from {5e-5, 5e-4, 1e-4, 1e-5}) | 0.2 (chosen from {0.1, 0.5, 0.2, 0.01, 0.0}) |
| Entity30 | 40 | 256 | 5e-5 (chosen from {5e-5, 5e-4, 1e-4, 1e-5}) | 0.2 (chosen from {0.1, 0.5, 0.2, 0.01, 0.0}) |
| Living17 | 40 | 256 | 5e-5 (chosen from {5e-5, 5e-4, 1e-4, 1e-5}) | 0.2 (chosen from {0.1, 0.5, 0.2, 0.01, 0.0}) |
| Nonliving26 | 40 | 256 | 0 5e-5 (chosen from {5e-5, 5e-4, 1e-4, 1e-5}) | 0.2 (chosen from {0.1, 0.5, 0.2, 0.01, 0.0}) |
| Officehome | 50 | 96 | 0.0001 (chosen from {0.0001, 0.001,1e-5, 0.0}) | 0.01 (chosen from {0.001, 0.01, 0.0001}) |
| DomainNet | 15 | 96 | 0.0001 (chosen from {0.0001, 0.001,1e-5, 0.0}) | 0.01 (chosen from {0.001, 0.01, 0.0001}) |
| Visda | 10 | 96 | 0.0001 (chosen from {0.0001, 0.001,1e-5, 0.0}) | 0.01 (chosen from {0.001, 0.01, 0.0001}) |
| Civilcomments | 5 | 32 | 0.01 (chosen from {0.01, 0.001, 0.0001, 0.0}) | 2e-5 (chosen from {$2e-4, 2e-5$}) |
| Retiring Adults | 50 | 200 | 0.0001 (chosen from {0.01, 0.001, 0.0001, 0.0}) | 0.01 (chosen from {0.001, 0.01, 0.0001}) |
| Mimic Readmissions | 100 | 128 | 0.0 (chosen from {0.01, 0.001, 0.0001, 0.0}) | 5e-4 (chosen from {0.005, 0.00010.0005}) |

Table 6: Details of the learning rate and batch size considered in our RLSBENCH

- **DANN, CDANN, IW-CDANN and IW-DANN**   As per Transfer Learning Library suggestion, we use a learning rate multiplier of $0.1$ for the featurizer when initializing with a pre-trained network and $1.0$ otherwise. We default to a penalty weight of $1.0$ for all datasets with pre-trained initialization.

- **FixMatch**   We use the lambda is $1.0$ and use threshold $\tau$ as $0.9$.

- **NoisyStudent**   We repeat the procedure for $2$ iterations and use a drop level of $p = 0.5$.

- **SENTRY**   We use $\lambda_{\text{src}} = 1.0$, $\lambda_{\text{ent}} = 1.0$, and $\lambda_{\text{unsup}} = 0.1$. We use a committee of size $3$.

- **PsuedoLabel**   We use the lambda is $1.0$ and use threshold $\tau$ as $0.9$.

### L.3   COMPUTE INFRASTRUCTURE

Our experiments were performed across a combination of Nvidia T4, A6000, P100 and V100 GPUs. Overall, to run the entire RLSBENCH suite on a T4 GPU machine with 8 CPU cores we would approximately need $70k$ GPU hours of compute.

### L.4   DATA AUGMENTATION

In our experiments, we leverage data augmentation techniques that encourage robustness to some variations between domains for vision datasets.

For weak augmentation, we leverage random horizontal flips and random crops of pre-defined size. For strong augmentation, we apply the following transformations sequentially: random horizontal flips, random crops of pre-defined size, augmentation with Cutout (DeVries & Taylor, 2017), and RandAugment (Cubuk et al., 2020). For the exact implementation of RandAugment, we directly use the implementation of Sohn et al. (2020). The pool of operations includes: autocontrast, brightness, color jitter, contrast, equalize, posterize, rotation, sharpness, horizontal and vertical shearing, solarize, and horizontal and vertical translations. We apply N = 2 random operations for all experiments.

## M   COMPARISON WITH SENTRY ON OFFICEHOME DATASET WITH DIFFERENT HYPERPARAMETERS

In this section, we shed more light on the discrepancy observed between SENTRY results reported in the original paper (Prabhu et al., 2021) and our implementation.

We note that for the main experiments on Officehome dataset, we used a batch size of 96 for all methods including SENTRY. However, SENTRY reported results with a batch size of 16 in their work. Hence, we re-run the SENTRY algorithm with a batch size of 16. To investigate the impact of

the decreased batch size, we make a comparison with FixMatch (the best algorithm on Officehome in our runs) by re-running it with the decreased batch size.

In Table 7 we report results on individual shift pairs in officehome. We observe that SENTRY improves over FixMatch for the default minor shift in the label distribution in the officehome dataset. However, as the shift severity increases we observe that SENTRY performance degrades. Overall, we observe that RS-FixMatch performs similar or superior to SENTRY on 3 out of 4 shift pairs in officehome.

| Algorithm | Alpha = None | Alpha = 10.0 | Alpha = 3.0 | Alpha = 1.0 | Alpha = 0.5 | Avg |
|---|---|---|---|---|---|---|
| FixMatch | 92.5 | 95.2 | 98.0 | 100.0 | 100.0 | 97.1 |
| RS-FixMatch | 92.5 | 96.4 | 98.0 | 100.0 | 100.0 | 97.4 |
| SENTRY | 93.0 | 94.0 | 98.0 | 83.3 | 87.5 | 91.2 |

(a) Product to Product (in-distribution)

| Algorithm | Alpha = None | Alpha = 10.0 | Alpha = 3.0 | Alpha = 1.0 | Alpha = 0.5 | Avg |
|---|---|---|---|---|---|---|
| FixMatch | 71.4 | 71.5 | 70.7 | 73.1 | 75.5 | 72.4 |
| RS-FixMatch | 74.7 | 74.0 | 72.1 | 73.1 | 70.4 | 72.9 |
| SENTRY | 78.1 | 78.0 | 75.1 | 71.7 | 65.3 | 73.6 |

(b) Product to Real

| Algorithm | Alpha = None | Alpha = 10.0 | Alpha = 3.0 | Alpha = 1.0 | Alpha = 0.5 | Avg |
|---|---|---|---|---|---|---|
| FixMatch | 41.5 | 44.0 | 44.2 | 48.4 | 39.4 | 43.5 |
| RS-FixMatch | 45.5 | 44.8 | 43.6 | 50.0 | 37.4 | 44.2 |
| SENTRY | 45.8 | 46.5 | 41.4 | 40.3 | 27.3 | 40.3 |

(c) Product to ClipArt

| Algorithm | Alpha = None | Alpha = 10.0 | Alpha = 3.0 | Alpha = 1.0 | Alpha = 0.5 | Avg |
|---|---|---|---|---|---|---|
| FixMatch | 54.4 | 51.3 | 54.7 | 57.3 | 55.9 | 54.7 |
| RS-FixMatch | 57.2 | 53.6 | 55.9 | 57.3 | 58.8 | 56.6 |
| SENTRY | 63.7 | 62.0 | 62.1 | 65.3 | 55.9 | 61.8 |

(d) Product to Art

Table 7: Officehome results with batch size 16 instead of 96 used throughout our experiments.

More generally, across our runs, we also observed model training with SENTRY to be unstable. Investigating further, we observe that the maximization objective to enforce consistency cause instabilities. This behavior is specifically prevalent for experiments where we don't use initiale the underlying model with pre-trained weights (for example, in BREEDs datasets).

