# OpenReview forum: "RLSBench: A Large-Scale Empirical Study of Domain Adaptation Under Relaxed Label Shift"
_ICLR.cc/2023/Conference — Submitted to ICLR 2023_

### Official Review · Reviewer_kXyy · 2022-10-21

**Confidence:** 5
**Correctness:** 3
**Technical Novelty And Significance:** 2
**Empirical Novelty And Significance:** 3
**Recommendation:** 6

**Clarity, Quality, Novelty And Reproducibility:**

Overall, this paper is of decent quality, introducing a comprehensive benchmark for an under-explored domain adaptation setting. The paper is generally presented clearly through there are some presentation errors and details that should be elaborated on. The authors provide detailed implementations that can be reproduced, and the results are in line with the result logs provided by the authors.

**Strength And Weaknesses:**

It is true that comprehensive and fair benchmarking of domain adaptation methods, especially in an under-explored but more realistic setting such as relaxed label shift. Therefore, the motivation of this paper is solid which investigates the effectiveness of prior domain adaptation methods under relaxed label shift in a comprehensive and fair manner. The experiments conducted are comprehensive and the observations of which generally support the authors' claims. The details of the implementation of the experiments are provided in a meticulous manner, with the rather detailed results provided in an anonymous GitHub form. These details greatly help the reproduction of the results and solidify the presentation of experiments. Overall, this paper deals with a domain adaptation setting of great research interest with comprehensive empirical results.

Despite the relatively solid motivation and comprehensive empirical results, there are a few weaknesses that the authors would consider addressing:
a) Even though the authors have proposed a set of meta-algorithm such that it could be leveraged with any domain adaptation (DA) methods and improve the capability of the DA methods in tackling label shift. However, the strategies used (i.e., re-sampling and re-weighting) are not novel techniques, as have also been acknowledged in the paper. Therefore, it would be interesting to analyze why such techniques, though useful, have not been considered in past literature. Additionally, while these strategies may not be leveraged previously on DA with relaxed label shift, they have been leveraged for other tasks with solid proof. The authors should therefore not only show that such methods work empirically, but also showcase why such methods work by providing theoretical proofs or theorems. This would enable a better understanding of why such methods should be employed for DA with relaxed label shifts.
b) Another concern about the proposed algorithm lies in the authors' claims (on Page 5, the "Summary" paragraph): "we expect DA methods to adapt classifier $f$ to $p(x|y)$ shift". This seems to contradict what the authors have stated in their review of previous domain adaptation methods in Section 2.1, where the authors noted that: "a few papers highlighted the theoretical and empirical failure of DA methods" for relaxed label shift setting. If so, it seems that the algorithm is only effective due to perhaps the meta-algorithm is able to adapt $f$ to the shift in $p(y)$. Can the authors elaborate on this contradiction?
c) While the authors describe in detail their observation through comprehensive sets of empirical results, the analysis behind these observations fall short of being satisfactory. For example, on Page 8 the authors observe that "early stopping criterion" and "data augmentation" would impose certain effects on the final result. However, other than a short description no further analysis of why these effects are imposed and to what degree subsequent methods should pay attention to these two aspects.
d) It is noted that the authors seem to take a different source/target domain setting approach than how certain domain adaptation benchmarks are normally leveraged. One example is the Office-Home dataset which normally benchmarks DA methods by setting 12 pairs of source/target domains with each domain being set as both the source and target domain. In this paper, though only the "Product" domain is set as the source domain. Could the authors prove that such settings would not affect the conclusions obtained? Meanwhile, to my understanding, the authors provide the average scores across all source/target domain pairs. Are there any outlier source-target domain pairs whose results are notably different than the rest? If there are, they should be discussed separately.
e) There are some typos and sentences that are not fluent enough, hampering the clarity of the presentation. The authors may check through their papers for these minor but sometimes obvious mistakes.

**Summary Of The Paper:**

This paper introduces a benchmark for evaluating domain adaptation methods under the relaxed label shift setting. The authors leverage the introduced benchmark on various domain adaptation methods showcasing the inability of prior domain adaptation methods in coping with the relaxed label shift setting. In addition, this paper also proposes a set of meta-algorithm that could bring general improvements to existing domain adaptation methods.

**Summary Of The Review:**

Generally speaking, this paper introduces a comprehensive benchmark that is of good significance towards the development of domain adaptation methods that are robust for real-world applications. The technical novelty of the paper may not be outstanding, but the empirical results do shed light on how relaxed label shift can be tackled if the authors could further elaborate in their analysis. I recommend a "marginally above the acceptance threshold" score.

---

> ### Author Response · Authors · 2022-11-17
> **Response to Reviewer kXyy**
>
> Thank you for your positive assessment and constructive feedback on our work. We are glad that you find the motivation for the problem studied in our work solid and our experimental results comprehensive.
>
>
> > **a) The strategies used (i.e., re-sampling and re-weighting) are not novel techniques, as have also been acknowledged in the paper. Therefore, it would be interesting to analyze why such techniques, though useful, have not been considered in past literature. Additionally, while these strategies may not be leveraged previously on DA … they have been leveraged for other tasks with solid proof. The authors should therefore not only show that such methods work empirically, but also showcase … by providing theoretical proofs or theorems. **
>
> The absence of comparison to re-sampling and re-weighting techniques in prior literature has been the main motivation for including these techniques in our study. The primary goal of our paper is to straighten up the current state of the empirical research, providing a common testbed for evaluating domain adaptation methods in relaxed label shift scenarios, and this contribution is vital given the lack of apples-to-apples comparisons in the current literature. Moreover,  one result of our extensive evaluation is to reveal a simple and effective method that has been missed in all the prior comparisons.
>
>
> We agree with the reviewers' suggestion about theoretically investigating re-sampling and re-weighting strategies in relaxed label shift scenarios. However, to the best of our knowledge,  most of the analysis for re-sampling is aimed at scenarios with im-balanced datasets where the true labels are known. We believe that formalizing a realistic model amenable to theoretical analysis of re-sampling with pseudolabels is definitely an interesting venue for future research.
>
>
> > **b) Another concern about the proposed algorithm lies in the authors' claims (on Page 5, the "Summary" paragraph): "we expect DA methods to adapt classifier to shift". This seems to contradict … authors stated … in Section 2.1, … If so, it seems that the algorithm is only effective due to perhaps the meta-algorithm is able to adapt to the shift in . Can the authors elaborate on this contradiction?**
>
> We apologize for the confusion here. The theoretical and empirical failure of domain alignment methods highlighted in the prior work is without any re-sampling step. Indeed, we show that without any re-sampling (step 1 of our meta-algorithm), popular DA methods tend to falter (Figure 1).  As correctly pointed out by the reviewer, the re-sampling step (to balance source and pseudo-balance target) is aimed to tackle the imbalance in target data  (with respect to
> the source label marginal). Our experimental results in Figure 2 corroborate this intuition where we observe that with increasing shift severity of target marginal, re-sampling improves the performance of existing DA techniques.
>
> To clarify the contradiction raised by the reviewer, we want to emphasize that we are hypothesizing the following: *we expect DA methods to adapt f to shift in p(x|y) under the condition that the re-sampling step in our meta-algorithm perfectly corrects for label marginal shift between source and target.* We have improved our exposition on this in the paper.
>
>
> > **c) While the authors describe in detail their observations through comprehensive sets of empirical results, the analysis behind these observations falls short of being satisfactory. For example, on Page 8 the authors observe that "early stopping criterion" and "data augmentation" would impose certain effects on the final result. However, other than a short description no further analysis of why these effects are imposed and to what degree subsequent methods should pay attention to these two aspects.**
>
> We thank the reviewer for this feedback. We have elaborated on our discussion from empirical findings in updated Section 4.
>
> Since (strong) data augmentation with vision datasets helps consistently, our results show that, whenever applicable, subsequent methods should use data augmentations. As discussed in our paper, in all of our implementations for vision datasets, we default to using strong data augmentations.
>
> For early stopping, we have added better illustrations and more discussion in Appendix H.  We observe a consistent $\approx$2\% and $\approx$8\% accuracy difference on vision datasets and tabular respectively with all methods, highlighting the importance of early stopping criteria. On NLP datasets, while the early stopping criteria have $\approx$2\% accuracy difference when RW and RS corrections are not employed, the difference becomes negligible when these corrections are employed. These results highlight that subsequent methods should clearly include a description of early stopping criteria.

---

> > ### Author Response · Authors · 2022-11-17
> > **Response to Reviewer kXyy (cont.)**
> >
> >
> > > **d) It is noted that the authors seem to take a different source/target domain setting approach…. One example is the Office-Home dataset which normally benchmarks DA methods by setting 12 pairs of source/target domains with each domain …  In this paper, though only the "Product" domain is set as the source domain. Could the authors prove that such settings would not affect the conclusions obtained?**
> >
> > In our initial experiments, we did follow the typical approach to permute over the different source and target pairs within a multi-domain dataset. We didn’t observe any differences in the key takeaways.
> >
> > Moreover, permuting across different source/target pairs for all multi-domain datasets scales the experiments by 3–4x  which becomes computationally very expensive at our scale (with 14 multi-domain datasets and a total of 560 different source/target pairs). Instead, since the kinds of natural variations in $p(x|y)$ that appear in the wild lack a rigorous characterization and often methods that work well on one type of shift need not generalize to others (Taori et al., 2020; Djolonga et al., 2020;), we experimented on a wider array of shifts by fixing the source set typically to the dataset with the largest number of labeled examples.
> >
> > > **Meanwhile, to my understanding, the authors provide the average scores … Are there any outlier source-target domain pairs whose results are notably different than the rest?**
> >
> > Thanks for this feedback. We have now updated all the results with plots illustrating each of our findings in Section 4. In particular, we have included box plots which now provide a more nuanced behavior (e.g., quartile, outlier, and variance information) than just mean accuracy numbers previously reported. We have also added detailed results on each dataset in Appendix I.
> >
> > > **e) There are some typos and sentences that are not fluent enough, hampering the clarity of the presentation.**
> >
> > Thanks for your suggestion. We have made a pass over the draft and fixed typographical errors.

---

> > ### Comment · Reviewer_kXyy · 2022-11-19
> > **Respond to the response to my reviews**
> >
> > I would like to thank the authors for their effort in addressing my concerns and questions. I believe the authors have addressed them well and have clarified my doubts. I personally insist that this paper is of good significance towards the development of domain adaptation methods that are robust for real-world applications, and therefore would really much vote positively for this paper to be accepted.

---

> > > ### Author Response · Authors · 2022-11-20
> > > **Thanks for your reply**
> > >
> > > Thanks for your reply and for championing acceptance. Please let us know if our updates are sufficient to justify increasing your score or if there are any other concerns that we can address.

---

### Official Review · Reviewer_mTKd · 2022-10-23

**Confidence:** 3
**Correctness:** 3
**Technical Novelty And Significance:** 2
**Empirical Novelty And Significance:** 2
**Recommendation:** 5

**Clarity, Quality, Novelty And Reproducibility:**

The writing of the paper is mostly clear. The method is easy to follow, and the codes are claimed to be published.

**Strength And Weaknesses:**

1. Strength: This paper benchmarks DA considering the target label marginal mismatch in relaxed label shift setting. It compares different DA methods with the proposed reweighting method on classifiers and resampling method for balancing the target data label distribution. The proposed two techniques can commonly improve DA methods in the relaxed label shift setting.

2. Weakness

I overall appreciate the benchmark in relaxed label shift setting, and the extensive comparisons, especially with the resampling and reweighting techniques. However, my major concerns on this work are the limited novelty.

(1) The benchmark datasets are all based on the previous popular DA datasets. The work of this paper is to resample the target domain data based on Dirichlet distribution to simulate the target label marginal shift.

(2) To handle the label marginal shift, the proposed RS (resampling based on pseudo-label) and RW (reweighting classifiers) are straightforward. These two techniques are combined with the other SoTA DA methods. This is reasonable, but the novel contribution in methodology for DA is limited.

(3) The RS and RW depend on the estimation of target domain label distribution, which are based on the pseudo-labels. The analysis of the accuracy of pseudo-labels and their effects on the DA performance need to be deepened.  From Table 3, it seems that the RS do not improve the target marginal estimation significantly.

**Summary Of The Paper:**

This paper works on benchmarking the domain adaptation methods on datasets with target label marginal shift and category data distribution p(x|y) shift. The benchmark datasets are based on the current popular DA datasets with resampled target label marginal distribution by resampling target domain data. The paper proposed resampling and reweighting techniques upon the current DA methods on this benchmark, and shows improvement over DA methods commonly, in the relaxed label shift setting.

**Summary Of The Review:**

The paper builds a benchmark for DA with target label marginal shift and the per-category data distribution shift. The benchmark dataset is simply by sampling target domain data using the popular DA datasets. The proposed RS and RW are reasonable for remedying the target label distribution shift, however, may heavily rely on the quality of pseudo-labels, and with limited contribution in methodology for DA.

---

> ### Author Response · Authors · 2022-11-17
> **Response to Reviewer mTKd**
>
> We thank the reviewer for their thoughtful and detailed feedback.
>
> > **The benchmark datasets are all based on the previous popular DA datasets. The work of this paper is to resample the target domain data based on Dirichlet distribution to simulate the target label marginal shift.**
>
> We agree with the reviewer that in RLSbench, we simulate target label marginal shift with stratified sampling. However, the aim of RLSbench is to offer a standardized test bed for fair and unbiased comparison of DA methods in relaxed label shift scenarios. In particular, our benchmark consists of 14 multi-domain datasets across vision, tabular, and language modalities (previously 11 datasets across vision) spanning > 500 distribution shift pairs with different class proportions. We also implement 13 popular domain adaptation methods (with and without re-sampling correction) in our testbed.
>
> Our comparisons provide previously unknown findings summarized in the general response. On the other hand, previous works that focus on relaxed label shift problems typically restrict their experiments to DomainNet and Officehome datasets which include sketches and synthetic renderings. Since the kinds of natural variations in $p(x|y)$ that appear in the wild lack a rigorous characterization and often methods that work well on one type of shift need not generalize to others (Taori et al., 2020; Djolonga et al., 2020;), we believe that it becomes crucial to test existing methods on a wider array of shifts.
>
>
> > **To handle the label marginal shift, the proposed RS (resampling based on pseudo-label) and RW (reweighting classifiers) are straightforward. These two techniques are combined with the other SoTA DA methods. This is reasonable, but the novel contribution in methodology for DA is limited.**
>
> As we acknowledge in our manuscript, we agree with the reviewer that re-sampling and re-weighting corrections leveraged in our meta-algorithm exist in prior work. The primary goal of our paper is not to introduce novel methods but to straighten up the current state of the empirical research, providing a common testbed for evaluating domain adaptation methods in relaxed label shift scenarios, and this contribution is vital given the lack of apples-to-apples comparisons in the current literature. Moreover, one result of our extensive evaluation is to reveal a simple and effective method that has been missed in all the prior comparisons.
>
> To the best of our knowledge, empirical comparison of these corrections is largely missing for relaxed label shift problems.  As a result of our extensive evaluation, we observe a simple and effective baseline that has been missed in all the prior comparisons. In particular, our experiments reveal a previously unknown finding: *Re-sampling and re-weighting corrections (i.e., our meta-algorithm) when paired with existing DA methods often improve significantly over the methods specifically proposed for the relaxed label shift settings (e.g., IWDANN, IWCDAN, SENTRY, etc).*
>
>
> > **The RS and RW depend on the estimation of target domain label distribution, which are based on the pseudo-labels. The analysis of the accuracy of pseudo-labels and their effects on the DA performance need to be deepened. From Table 3, it seems that the RS do not improve the target marginal estimation significantly.**
>
> We have improved the exposition of our main results in Section 4 for discussion on RS and RW corrections. Appendix G includes detailed results.
>
> Regarding improvement for target marginal estimation, we want to make a clarification. Our main observation that estimators leveraging DA classifiers tend to perform better than using source-only classifiers for tabular and vision datasets. Figure 3 shows our results. More results are in App G. Benefits with RS steps comes mainly when the target marginal shift is severe (i.e., $\alpha \in$ \{1.0, 0.5\}).

---

> > ### Author Response · Authors · 2022-12-01
> > **Are there any additional questions or concerns?**
> >
> > We hope that our responses above have addressed the concerns raised in the review. **Are there any additional concerns? We would be happy to continue the discussion if there are additional questions or concerns.**

---

### Official Review · Reviewer_PHqQ · 2022-10-24

**Confidence:** 4
**Correctness:** 3
**Technical Novelty And Significance:** 2
**Empirical Novelty And Significance:** 4
**Recommendation:** 6

**Clarity, Quality, Novelty And Reproducibility:**

Clarity: Good, this paper is well-written.

Quality: Good. Experiments are comprehensive.

Novelty: Limited. But the contribution is not marginal.


**Details Of Ethics Concerns:**

No.

**Strength And Weaknesses:**

Strengths:
+ This paper is well-written. It is enjoyable to read.
+ This paper studies an important problem. While existing DA approaches work well under covariance shift and label shift. More general scenarios should be introduced to test the effectiveness of the proposed method.
+ Experiments results are self-contained, and many insightful observations and discussions are provided.

Weaknesses:
+ It can be observed that some semi-supervised learning methods (e.g. Fixmatch) work well in relaxed label shift settings. Fixmatch is a consistency-based method. In experiments we find that adversarial learning based methods show worse performance than consistency-based methods. In DA, MCD[1] is an adversarial learning based method. And it is also a consistency-based method. What about this method in the benchmark?
[1] Maximum classifier discrepancy for unsupervised domain adaptation
+ More insightful analysis should be provided. For example, why Fix-match works better than other methods?
+ It would be interesting if more benchmarks (more than vision datasets, for example, NLP?) can be introduced.

**Summary Of The Paper:**

This paper studies popular domain adaptation methods under scenarios where both label distribution and conditionals may shift. Specifically, the authors introduce a large-scale benchmark for relaxed label shift settings which consists of 11 vision datasets spanning more than 200 distribution shift pairs. 12 popular domain adaptation methods are evaluated on this benchmark and several observations are provided. This paper shed light on domain adaptation methods in relaxed label shift settings.

**Summary Of The Review:**

Strengths:
+ This paper is well-written. It is enjoyable to read.
+ This paper studies an important problem. While existing DA approaches work well under covariance shift and label shift. More general scenarios should be introduced to test the effectiveness of the proposed method.
+ Experiments results are self-contained, and many insightful observations and discussions are provided.

Weaknesses:
+ It can be observed that some semi-supervised learning methods (e.g. Fixmatch) work well in relaxed label shift settings. Fixmatch is a consistency-based method. In experiments we find that adversarial learning based methods show worse performance than consistency-based methods. In DA, MCD[1] is an adversarial learning based method. And it is also a consistency-based method. What about this method in the benchmark?
[1] Maximum classifier discrepancy for unsupervised domain adaptation
+ More insightful analysis should be provided. For example, why Fix-match works better than other methods?
+ It would be interesting if more benchmarks (more than vision datasets, for example, NLP?) can be introduced.

Overall, the authors should give more insightful discussions to further improve the paper. And this paper can shed light on DA in relaxed label shift setups.

---

> ### Author Response · Authors · 2022-11-17
> **Response to Reviewer PHqQ**
>
> Thank you for your positive assessment and constructive feedback on our work. We are glad that you find the problem studied in our work important and that our findings/observations are insightful.
>
>
> > **It would be interesting if more benchmarks (more than vision datasets, for example, NLP?) can be introduced.**
>
> Thanks for this suggestion. We have now added included NLP and tabular datasets expanding our benchmark beyond vision datasets. In particular, for tabular, we have now included Retiring Adults and MIMIC Readmission multi-domain datasets. For NLP, we have included the Civilcomments dataset. Overall, we obtain additional 120 different distribution shift pairs of varying severity of class proportion shift.
>
> Our observations continue on hold on new modalities. In particular, our findings about (i) the susceptibility of DA methods to failure under extreme shifts in the class proportions; and (ii) the effectiveness of our meta-algorithm when paired with existing DA techniques continue to hold (in a more pronounced way) on tabular and NLP datasets. Refer to results in App B.
>
> > **It can be observed that …  Fixmatch work well in relaxed label shift settings …. In DA, MCD[1] is an adversarial learning based method. And it is also a consistency-based method. What about this method in the benchmark?**
>
> Thanks for this reference. We have added an implementation of this method in our benchmark by referring to implementation [here](https://github.com/mil-tokyo/MCD_DA) and we are running experiments with this method.
>
> Since re-running this method across all the distribution shift pairs is computationally costly, we have added results on a selected subset (that finished early) below. We will keep updating the paper as we get more results.
>
> | Dataset    | DANN  | CDANN | MCD   | FixMatch |
> |------------|-------|-------|-------|----------|
> | CIFAR10    | 90.86 | 90.00 | 89.03 | 91.87    |
> | CIFAR100   | 74.80 | 74.57 | 73.22 | 79.03    |
> | Officehome | 66.05 | 66.19 | 64.91 | 65.48    |
>
>
>
> > **More insightful analysis should be provided. For example, why Fix-match works better than other methods?**
>
> We agree with the reviewer that more analysis is needed to understand why methods like FixMatch work better. Several recent papers have been trying to understand the effectiveness of consistency regularization objectives in a population sense [1,2] and more work is required to better understand why iterative self-training with consistency regularization helps.
>
> Our work in this regard highlights the susceptibility of DA methods (e.g. FixMatch) to failure under extreme shifts in the class proportions. Moreover, we show that simple re-sampling and re-weighting correction can often alleviate this susceptibility. We believe that theoretically characterizing this behavior of FixMatch is beyond the scope of the current paper. But we would be more than happy to include additional analysis if the reviewer has specific suggestions for us.
>
>
> [1] Theoretical Analysis of Self-Training with Deep Networks on Unlabeled Data. Colin Wei, Kendrick Shen, Yining Chen, Tengyu Ma. ICLR 2021.
>
> [2] Cai, Tianle, Ruiqi Gao, Jason Lee, and Qi Lei. A theory of label propagation for subpopulation shift. In ICML 2021.

---

> > ### Author Response · Authors · 2022-12-01
> > **Are there any additional questions or concerns?**
> >
> > We hope that our responses above have addressed the concerns raised in the review. **Are there any additional concerns? We would be happy to continue the discussion if there are additional questions or concerns.**

---

### Official Review · Reviewer_VEqd · 2022-10-24

**Confidence:** 5
**Correctness:** 3
**Technical Novelty And Significance:** 2
**Empirical Novelty And Significance:** 2
**Recommendation:** 3

**Clarity, Quality, Novelty And Reproducibility:**

The paper is well-written and easy to follow. But there lacks of inspirational conclusions from the experimental results. The novelty is limited.

**Strength And Weaknesses:**

Strength:
1) This paper gives a large-scale evaluation of existing domain adaptation algorithms under relaxed label shift scenarios. This helps related researchers focus on this problem.
2) Commonly adopted datasets and many related methods are conducted.

Weakness:
1) This paper evaluated domain adaptation methods on relaxed label shift scenarios. But there is no formal definition of the relaxed label shift.
2) The authors only report the mean accuracy of different methods. How about the variance?
3) The observation from the experimental results all seems to be well-known conclusions. Are there some inspirational or unusual conclusions?


**Summary Of The Paper:**

This paper gives the empirical study of domain adaptation algorithms under relaxed label shift scenarios. Many algorithms such as domain-invariant representation learning, self-training, and test-time adaptation methods across 11 multi-domain datasets are conducted.

**Summary Of The Review:**

Based on the above discussion, I tend to reject.

---

> ### Author Response · Authors · 2022-11-17
> **Response to Reviewer VEqd**
>
> We thank the reviewer for their consideration of our paper.
>
> > **This paper evaluated domain adaptation methods on relaxed label shift scenarios. But there is no formal definition of the relaxed label shift.**
>
> We have added a definition in the paper and improved the exposition as per your comment:
>
> Domain adaptation problems are, in general, ill-posed. The label-shift assumption, where $p(x|y)$ does not change but that $p(y)$ can, makes the problem well-posed. However, these assumptions are typically, to some degree, violated in practice. Our paper aims to relax this assumption and focuses on *relaxed label shift* setting. In particular, we assume that the label distribution can shift from source to target arbitrarily but that $p(x|y)$ varies between source and target in some comparatively restrictive way, excluding settings where p(x|y) can shift arbitrarily. One such example is shifts arising naturally in the real world like ImageNet to ImageNetV2.
>
> Mathematically, we assume a divergence-based restriction on  $p(x|y)$, i.e., for some small $\epsilon > 0$ and distributional distance $D$, we have  $\max_y D(p_t(x|y), p_t(x|y)) \le \epsilon$ but allowing an arbitrary shift in the label marginal $p(y)$. We can instantiate the divergence with Wasserstein-infinity distance to define our constraint, i.e.,  $D = \max_y W_\infty(p_s(x|y), p_t(x|y)) \le \epsilon$. We can also define our distribution constraint in KL or TV distances. Refer to App F for more discussion and formal definitions.
>
> However, in practice, it's hard to empirically verify these distribution distances for small enough $\epsilon$ with finite samples. Moreover, we lack a rigorous characterization of the sense in which those shifts arise in popular DA benchmarks, and since, the focus of our work is on the empirical evaluation with real-world datasets, we leave a formal investigation for future work.
>
>
>
> > **The authors only report the mean accuracy of different methods. How about the variance?**
>
> Thanks for this feedback. We have now updated all the results with plots illustrating each of our findings in Section 4. In particular, we have included box plots which now provide a more nuanced behavior (e.g., quartile, outlier, and variance information) than just mean accuracy numbers previously reported.
>
>
> > **The observation from the experimental results all seems to be well-known conclusions. Are there some inspirational or unusual conclusions?**
>
> We have improved exposition in our Introducition and Section 4 (i.e., Main Results) to highlight salient findings.
>
> The primary goal of our paper is to straighten up the current state of the empirical research in relaxed label shift scenarios and highlight a simple and effective method that has been missed in all the prior comparisons. To the best of our knowledge, while re-sampling and re-weighting corrections leveraged in our meta-algorithm exist in prior work, empirical comparison of these corrections is largely missing for relaxed label shift problems. In fact, our extensive experiments reveal a previously unknown finding: Re-sampling and re-weighting corrections (i.e., our meta-algorithm) when paired with existing DA methods often improve significantly over the methods specifically proposed for the relaxed label shift settings (e.g., IWDANN, IWCDAN, SENTRY, etc) when compared in a fair and comprehensive manner.
>
> If these findings are previously known, we would appreciate references to related work that we have missed.
>
> Overall, our work provides: (i) extensive empirical comparisons that underscore the importance of a fair comparison to avoid a false sense of scientific progress in relaxed label shift scenarios; (ii) RLSbench, a standardized framework for rigorous and reproducible research in the field; and (iii) a meta-algorithm, which when paired with existing DA techniques, can act as a strong baseline in future comparisons.

---

> > ### Author Response · Authors · 2022-12-01
> > **Are there any additional questions or concerns?**
> >
> > We hope that our responses above have addressed the concerns raised in the review. **Are there any additional concerns? We would be happy to continue the discussion if there are additional questions or concerns.**

---

### Author Response · Authors · 2022-11-17
**General Response**

We would like to thank the reviewers for their thoughtful feedback. We are glad to see that reviewers appreciated our comprehensive benchmark (PHqQ, kXyy), large-scale evaluation (VEqd, PHqQ, mTkd, kXyy) and that the paper is well-written (PHqQ, mTkd, kXyy)).
While the initial assessment of the paper includes mixed scores, we are optimistic that we can address key concerns with the updated draft and our responses.

Per reviewers' feedback, we have made several significant improvements to the paper which we believe strengthens its contributions. Summary of key changes in the draft:

- Per reviewer PHqQ’s suggestion, we have extended our benchmark to include **several NLP and tabular datasets** (previously, we only focused on problems in computer vision). Among tabular datasets, we have included Retiring Adults [1] and MIMIC Readmission [2] multi-domain datasets. For NLP, we have included the Civilcomments dataset [3].
Repeating experiments across these three datasets, we observe the same susceptibility of DA methods to failure under extreme shifts in the class proportions. In particular, the performance of classifiers obtained with existing DA techniques (e.g. DANN, Pseudolabel) falls below the performance of vanilla source classifiers (e.g., ERM on labeled source data). Moreover, our meta-algorithm (consisting of re-sampling and re-weighting steps) alleviates the aforementioned susceptibility of DA methods, often significantly improving over vanilla source classifiers.

- We have now run two seeds for each alpha, overall obtaining 560 different distribution shift pairs. We have updated the draft significantly to **improve the exposition with better illustrations** of our findings. In particular,

  - Added plots in the paper illustrating each of our findings in Section 4. In particular, we have included box plots which now provide a more nuanced behavior (e.g., quartile, outlier, and variance information) than just mean accuracy numbers previously reported.
  - We have added a more elaborate discussion of each of our takeaways in Section 4.


Finally, we would like to clarify our main contributions:

1. We present RLSbench, a standardized benchmark for settings with a shift in class proportion and natural variation in $p(x|y)$, consisting of 14 datasets across vision, tabular, and language modalities spanning > 500 distribution shift pairs with different class proportions. We also implement 13 popular domain adaptation methods in our testbed.


2. We demonstrate a widespread susceptibility of DA methods to failure under extreme shifts in the class proportions than was previously known.


3. We develop an effective meta-algorithm that consists of the following two steps: (i) pseudo-balance the data at each epoch; and (ii) adjust the final classifier with (an estimate of) target label distribution.


4. Our meta-algorithm, when paired with existing deep DA heuristics, often improves significantly over the methods specifically proposed for the relaxed label shift settings (e.g., IWDANN, IWCDAN, SENTRY, etc) when compared in a fair and comprehensive manner.


We believe that our open-source code and setup will foster reproducible and rigorous research in the field where our meta-algorithm (paired with existing DA methods) can act as strong baselines.

We respond to each reviewer in detail in their individual threads. In light of these new clarifications, please let us know if you have any remaining concerns. We are happy to answer any further questions you may have.

[1] Frances Ding, Moritz Hardt, John Miller, and Ludwig Schmidt. Retiring adult: New datasets for fair machine learning. Advances in Neural Information Processing Systems, 34:6478–6490, 2021.

[2] Alistair Johnson, Lucas Bulgarelli, Tom Pollard, Steven Horng, Leo Anthony Celi, and Roger Mark. Mimic-iv. PhysioNet. Available online at: https://physionet. org/content/mimiciv/1.0/(accessed August 23, 2021), 2020

[3] Daniel Borkan, Lucas Dixon, Jeffrey Sorensen, Nithum Thain, and Lucy Vasserman. Nuanced metrics for measuring unintended bias with real data for text classification. In Companion Proceedings of The 2019 World Wide Web Conference, 2019.

---

### Decision · Program_Chairs · 2023-01-20

**Decision:**

Reject

**Justification For Why Not Higher Score:**

N/A

**Justification For Why Not Lower Score:**

N/A

**Metareview: Summary, Strengths And Weaknesses:**

This paper consists in an empirical study of domain adaptation algorithms under relaxed label shift scenarios. An extensive experimental analysis over several datasets is carried out. This work also proposes a meta-learning algorithm consisting in re-sampling and re-weighting data samples which, when applied to standard DA methods, often improve performance wrt the methods specifically proposed for the relaxed label shift setting. The meta-learning method is not per se a strong contribution of the paper, as also acknowledged by the authors.
This work received contrasting evaluations/ratings, which also persisted after authors' rebuttal.
The main issue regards the fact that this paper presents a thorough empirical analysis without proposing any novel method to cope with this specific task. So, the originality level is rather weak, but the experimental analysis is quite comprehensive for the task at hand. So, the problem here is to decide if a paper of this type is enough valuable to be acceptable to ICLR conference.

Reviewers evidenced many remarks, e.g., other than the weak novelty as main issue, it was asked to better explaining the motivations and justifications of the work, improving the experimental analysis (in very many ways, e.g., more datasets, not only related to computer vision, more comparative analyses), better highlighting the actual takeaway message of this type of work, corroborating this study with a theoretical analysis, etc. Critics have also regarded how the proposed dataset was built, that is, as a merge of several public datasets corrupted artificially to simulate target label marginal shift,

Authors and reviewers were engaged in further discussions after the first round of reviews, authors replied extensively and in many cases satisfactorily, but still the paper was on the fence.

AC opinion is that an ICLR paper should have a significant original contribution in the methodology, but this was not the case.
AC recognizes that there is for sure a discordance on the way DA algorithms are validated, especially for the particular addressed task, but only providing an empirical evidence that this is the case is not sufficient. Further, the proposed benchmark dataset is not an actual dataset affected by a relaxed label shift, but this specific characteristic was artificially generated over other public "normal" benchmarks. Finally, the conclusions of this study are quite straightforward, they do not seem to disclose any relevant new finding.

In conclusion, considering all raised remarks, my decision is that the paper is not acceptable to ICLR23.
The work is surely valuable and, to some extent, useful for the community, but maybe it should be submitted to another venue or a journal, to have the possibility to better evidence actual contributions.




**Summary Of Ac-Reviewer Meeting:**

Just a couple of reviewers discussed separately with me, one remained negative even if was also open to acceptance, while another changed idea becoming positive. In the end, none was pretty sure of his/her evaluation, and indeed, this work can be accepted or rejected on the basis of the sensitivity one has about a work practically only empirical (even extensive), without a clear methodological novel contribution, and not providing any new conclusion.
In this scenario, I am slightly leaning towards rejection even if there can be reasons for acceptance too.